



# Surface nuclear magnetic resonance for studying an englacial channel on Rhonegletscher (Switzerland): Possibilities and limitations in a high-noise environment

Laura Gabriel [1,2], Marian Hertrich[3], Christophe Ogier[1,2], Mike Müller-Petke[4], Raphael Moser[1,2], Hansruedi Maurer[3], and Daniel Farinotti[1,2]

[1]Laboratory of Hydraulics, Hydrology and Glaciology (VAW), ETH Zurich, Zurich, Switzerland
[2]Swiss Federal Institute for Forest, Snow and Landscape Research (WSL), bâtiment ALPOLE, Sion, Switzerland
[3]Institute of Geophysics, ETH Zurich, Zurich, Switzerland
[4]Leibniz Institute for Applied Geophysics, Stilleweg 2, D 30655, Hannover, Germany

**Correspondence:** Laura Gabriel  (laura.gabriel@vaw.baug.ethz.ch)

**Abstract.** Surface nuclear magnetic resonance (SNMR) is a geophysical technique that is directly sensitive to liquid water. In this study, we evaluate the feasibility of SNMR for detecting and characterizing an englacial channel within Rhonegletscher, Switzerland. Building on prior information on Rhonegletscher's englacial hydrology, we conducted a proof-of-concept SNMR survey in the summer of 2023. Despite the high levels of electromagnetic noise, careful optimization of SNMR data processing

including remote reference noise cancellation, allowed us to successfully detect interpretable signals and to estimate parameters for a simplified one-dimensional water model. Our analysis, which is based on the comparison of the error-weighted root-mean-square misfit $\chi^{\mathrm{RMS}}$ of different models, suggests the existence of an aquifer near the bedrock, embedded within a temperate-ice column. Assuming a minimum aquifer water content of $60\,\%$, models with $\chi^{\mathrm{RMS}} \leq 1.9$ point to a thin layer ($\leq 1\,\mathrm{m}$) located at a depth of 44 to $60\,\mathrm{m}$, surrounded by temperate ice with a liquid water content between $0.3\,\%$ and $0.75\,\%$. Our findings

are consistent with complementary ground penetrating radar measurements and previous GPR studies, thereby corroborating the potential for using SNMR in englacial studies. Although limited by noise and model simplifications, our analyses show promise for quantifying liquid water volume located within or beneath glaciers.

## 1  Introduction

Glacial hydrology can be investigated with a number of experimental methods, ranging from direct observations via borehole

measurements to geophysical techniques. The latter are particularly relevant as they are non-invasive, and they have the potential to reveal the structure of large volumes of the glacier's subsurface. Active and passive seismic methods (e.g. Guillemot et al., 2024; Nanni et al., 2021; Lindner et al., 2020; Podolskiy and Walter, 2016; Peters et al., 2008) as well as ground-penetrating radar (GPR) (e.g. Church et al., 2021; Hansen et al., 2020; Irvine-Fynn et al., 2011; Moorman and Michel, 2000) are popular choices in this respect, and have been employed to study the location, geometry, water flow or temporal evolution of the en- and subglacial hydrological system. While GPR and seismics are effective at detecting the boundaries of englacial structures,

they do not provide direct information about water content in the ice, which can be of particular interest in the context of hazard



management, like in the case of glacier water pocket outburst floods (Ogier et al.; Vincent et al., 2012; Haeberli, 1983).
Surface nuclear magnetic resonance (SNMR), a geophysical method introduced in the 1980s (Schirov et al., 1991; Semenov et al., 1988), is a method directly sensitive to water molecules and, therefore, has the potential to directly reveal the water content of the subsurface. SNMR operates on principles similar to magnetic resonance imaging used in medical applications. When placed in a static magnetic field, such as Earth's geomagnetic field $B_{\text{earth}}$, the nuclear magnetic moments of the hydrogen atoms contained in the water molecules partially align with the static field and precess at the so-called Larmor frequency $f_{\text{L}}$. The latter is given by

$$f_{\text{L}} = \gamma B_{\text{earth}}/2\pi, \tag{1}$$

where $\gamma$ is the gyromagnetic ratio. The collective alignment of magnetic moments results in a net magnetic moment parallel to Earth's magnetic field, and when an additional magnetic field is applied in the form of a pulse oscillating at the Larmor frequency, the magnetic moments rotate out of their equilibrium configuration. As the magnetic moments relax back to equilibrium (typically characterized by the transverse relaxation time $T_2^*$ in SNMR experiments), they induce changes in the local magnetic field, which can be detected and used to infer information on the actual water content. In practice, the magnetic pulse is generated by an electrical current flowing through a large transmitter loop (up to 150 m in diameter), and measured by a similarly sized receiver loop. More information on the background of the technique can be found, e.g. in Hertrich (2008) or Weichman et al. (2000).

So far, cryospheric applications of SNMR are relatively limited: SNMR has been used in combination with GPR to characterize and estimate the volume contained in a glacier water pocket in the French Alps (Vincent et al., 2012; Legchenko et al., 2011). SNMR has also proven useful for detecting water in permafrost (e.g. Parsekian et al., 2019, 2013), sea ice (Nuber et al., 2013) or below a proglacial moraine (Lehmann-Horn et al., 2011), but in general, the applications are not widespread. One of the reasons is that SNMR surveys typically involve significant field efforts, which can be even more pronounced in areas with limited accessibility, like glaciers or sea ice. Loop placement and measurement durations can be time-consuming. On top of that, SNMR measurements often have low signal-to-noise ratios (S/N), necessitating multiple processing steps to extract meaningful information from the raw data. The latter is particularly limiting, when attempting to detect smaller water volumes in noisy environments, making the results uncertain.

In this study, we investigate the potential of SNMR for detecting an englacial channel in Rhonegletscher, Switzerland. For our study area, we expect a relatively poor S/N due to the comparatively small water volume in an englacial channel (small compared to e.g. the water pocket in Vincent et al. (2012)). Building on previous research that detected an englacial channel in the terminal part of Rhonegletscher (Church et al., 2021, 2020, 2019) and preliminary SNMR investigations conducted in the same area in 2008 (Hertrich and Walbrecker, 2008), we conduct a proof-of-concept study pursuing the following objectives: (1) Evaluate the detectability and possibility of characterizing Rhonegletscher's englacial channel with SNMR; (2) identify the specific challenges associated with such a survey, with a focus on the poor S/N; and (3) present future perspectives for applications of SNMR on mountain glaciers.



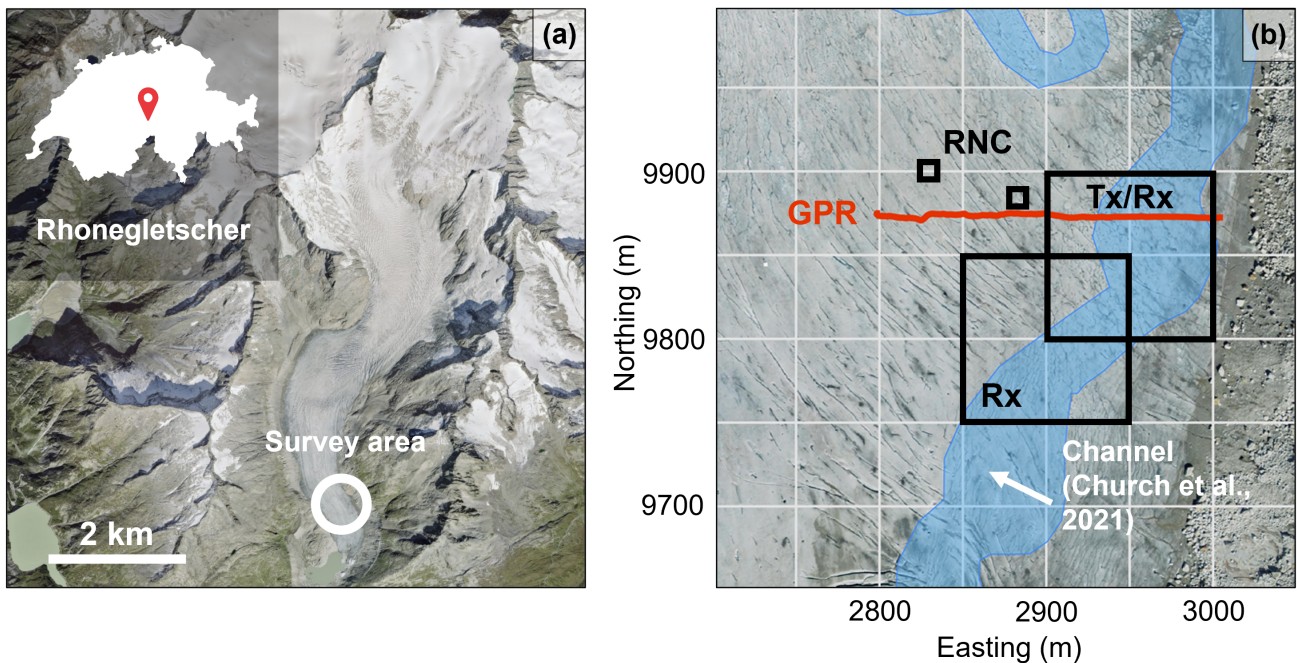

**Figure 1.** Overview of the survey site on Rhonegletscher. (a) Aerial view of Rhonegletscher in Central Switzerland. The white circle indicates the survey area. (b) Aerial view of the glacier tongue where we conducted the survey. The squares represent the different SNMR loops and depict the transmitter (Tx), receiver (Rx) and remote reference noise cancellation (RNC) loops. The estimated location of the englacial channel (Church et al., 2021) is shown in blue. The red line corresponds to the GPR profile shown in Figure 9. Coordinates are given in the CH1903+/LV95 system and are displayed with respect to an easting of 267000 m and a northing of 1150000 m. Orthophotos are provided by the Swiss Federal Office of Topology, © swisstopo 2023.

## 2 Study site and data

### 2.1 Site description

Rhonegletscher is a temperate glacier located at the East end of the Rhone valley in the canton of Valais, Switzerland (Fig. 1a). With a size of $16.4 \, \text{km}^2$ and a length of 9.7 km in 2016, it is one of the largest glaciers in Switzerland (GLAMOS, 2018). Between 2012 and 2020, Church et al. (2021, 2020, 2019) conducted borehole, seismic and GPR campaigns in the ablation zone of Rhonegletscher to enhance the understanding of the local englacial hydrology. The studies were able to reconstruct a three-dimensional model of the main en- and subglacial channel of the ablation zone (Church et al., 2021). Figure 1b shows a part of the extent of this channel, estimated from the 3D-GPR data acquired in the summer of 2020.

Motivated by their findings, we conducted a proof-of-concept study in the summer of 2023 investigating a section of the englacial channel with SNMR. We investigated the area presented in Figure 1b, corresponding to a portion of the area previously studied with GPR (Church et al., 2021, 2020, 2019). The survey area was constrained by the terrain, accessibility,





equipment and time. To validate the findings from the SNMR campaign, we complemented our work with a GPR survey at the same site (Sec. 2.2.2).

## 2.2 Data acquisition

### 2.2.1 SNMR survey

We conducted the SNMR field survey using a Numis Poly instrument manufactured by Iris Instruments (www.iris-instruments.com). Numis Poly belongs to the second generation of SNMR instruments (Dlugosch et al., 2011) offering four detection channels. We utilized the software called Prodiviner, provided by Iris Instruments, to control the measurement and acquire the time series.

For the survey, we deployed a total of four loops (Figure 1b). One loop is used as transmitter (Tx in Fig. 1b) and generates the pulsed magnetic field interacting with the water molecules. All four loops, including the transmitter loop, subsequently
record a voltage time series reflecting local changes in the magnetic field, comprising noise and the SNMR signal. Hereby, two loops are used as receivers (Rx in Fig. 1b) and two loops as so-called remote reference (RNC in Fig. 1b).

The two receiver loops measure the time series we use to extract the SNMR signal. One receiver loop corresponds to the transmitter loop (coincident-loop configuration), while the other receiver loop overlaps with the transmitter loop (separate-loop configuration). The latter arrangement can offer complementary information on the subsurface compared to the standard
coincident-loop setup (Hertrich et al., 2009). The optimal loop size depends on the desired depth of investigation and the resolution, larger loops offering greater penetration depths at the expense of spatial resolution (Kremer et al., 2022). In 2020, the depth of the channel in the survey area was estimated to be around 70 meters (Church et al., 2021), and we expect this depth to have decreased in 2023 due to surface melt. We thus deployed 100-meter single-turn square loops for both the receiver and transmitter, as we expect this size to offer the best compromise between penetration depth and spatial resolution.

The time series of the two remote reference noise cancellation loops (RNC loops in the following) are used to remove spatially correlated noise from the receiver time series, thereby enhancing the signal-to-noise ratio. Ideally, RNC loops should be placed at a distance of three times the Tx-loop diameter (center-to-center) to record time series that only comprise noise (Dlugosch et al., 2011). In our case, these loops were placed at a center-to-center distance between ∼80 and ∼120 meters from the transmitter loop (Fig. 1b), which makes contamination with SNMR signal likely (see discussion in Section 5.4.1). For the
RNC loops, we used a configuration suggested by Iris Instruments, which involved 10-meter square loops with seven turns (instruments, 2019).

In the acquisition software, we selected the following four SNMR-measurement parameters (cf. Tab. 1) and kept them constant for all acquisitions:

(1) The reference frequency $f_r$ was set to the local Larmor frequency $f_L$, which we estimated from the local geomagnetic
field (Eq. 1). For that, we measured the Earth's magnetic field using a Geometrics' G-858 Cesium vapour magnetometer, obtaining Larmor frequencies between 2039.1 and 2039.2 Hz. Given the temporal variations in Earth's magnetic field, the Larmor frequency undergoes small, continuous changes, the implications of which we discuss in Section 4.1.

(2) The pulse moment is obtained from $q = I\tau_p$, where $I$ is the excitation-pulse current amplitude and $\tau_p$ is the excitation-pulse



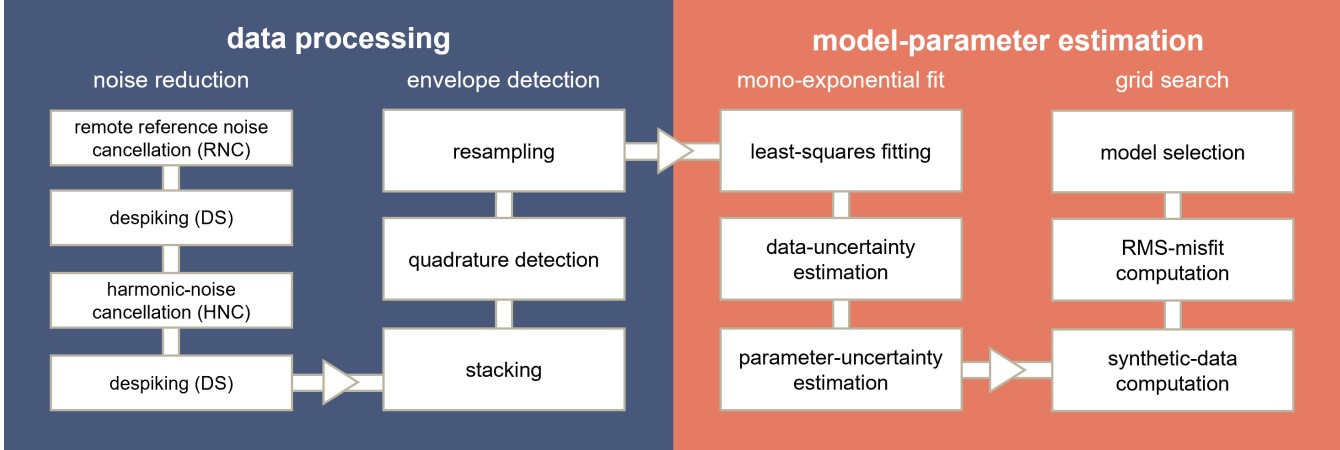

**Figure 2.** Schematic overview of the workflow entailing the data processing (blue) and the model-parameter estimation (orange). Details are found in Section 3.

duration, which we set to 40 ms. During one measurement series, we scanned through 16 different pulse moments by varying the current amplitude $I$ at a constant pulse duration. By increasing the pulse moment, we probed different volumes of the subsurface.

(3) The excitation pulse is followed by the so-called dead time $\tau_d$ (roughly 40 ms) before the recording of the time series starts. We chose the maximum recording time of 1.0 s since we expect relaxation times up to 1.5 s in pure liquid water (Grunewald and Knight, 2011; Schirov et al., 1991). Additionally, we recorded 1.0 s noise-only traces prior to each excitation pulse.

(4) For each pulse moment, we repeated the measurements 96 times (this number is called the "stacking number") to reduce the overall noise levels. The chosen stacking number is a compromise between measurement duration and noise reduction.

Based on the above measurement parameters, the survey encompassed $96 \cdot 16 = 1536$ single measurements. However, due to problems with the hardware, the measurement with $\#q = 16$ could not be completed and we only consider the measurements up to $\#q = 15$ in our analysis. The total measurement duration was almost seven hours, and we required a few additional hours to lay out the loops and set up the necessary equipment.

### 2.2.2 GPR survey

We acquired the GPR profile visible in Figure 1b using a Sensor & Software pulseEKKO Pro GPR system with antennas operating at a central frequency of 50 MHz. The system was equipped with a Leica real-time differential GNSS receiver to continuously track its position. We acquired the profiles by carrying the antennas at ca. 50 cm above the ground. The separation between the transmitter and receiver antenna amounted to 2 m. We processed all profiles with the in-house software GPRglaz (Grab et al., 2018) and following the standard processing workflow presented, for example, in Ogier et al. (2023); Grab et al. (2021); Church et al. (2020).





## 3  Methodology

To derive quantitative information about the glacier's englacial water content from the raw time series, we apply a four-step
procedure (Fig. 2). In a nutshell, this procedure entails a data-processing sequence including (1) noise reduction and (2) envelope detection, and a model-parameter estimation sequence including both a (3) mono-exponential fit and (4) grid search. These individual steps are described in more detail below. All steps are based on functionalities of the software "MRSmatlab" (Müller-Petke et al., 2016) version 2021.

### 3.1  Data processing

#### 3.1.1  Noise reduction

SNMR measurements often suffer from low S/N, requiring multiple data-processing steps to filter the noise. Figure 3a shows an exemplary raw time series of the data set obtained on Rhonegletscher, which is entirely dominated by noise (we discuss this noise and its potential sources in more detail in Sec. 5.3). If a clear SNMR signal was apparent, an oscillating decay should be visible. Since this is not the case, noise filtering was necessary.

MRSmatlab offers three noise-filtering approaches, each targeting different noise types.

(1) Despiking (DS) removes extreme values (so-called spikes), like the one reaching more than $10^5$ nV in Fig. 3a. Spikes are typically a result of powerful discharges like lightning. While we identify multiple spikes in the data sets acquired on Rhonegletscher, they do not dominate the overall noise.

(2) Harmonic Noise Cancellation (HNC) filters components of higher harmonics of anthropogenic, fundamental frequencies.
For instance, oscillations from power lines at 50 Hz can contaminate the signal near the Larmor frequency. On Rhonegletscher, we observe higher harmonics of $\approx 50$ Hz and $\approx 16.6$ Hz. However, their relative contribution to the total noise is minor. We had to choose a relatively large range of possible frequencies (16.45 - 16.85 Hz) to effectively cancel harmonic noise around 16.6 Hz.

(3) Remote Reference Noise Cancellation (RNC) targets the noise of unknown characteristics, which is dominating our data.
We deployed two remote reference loops to record the time series simultaneously with the two receiver loops (Fig. 1b). For this analysis, we only use the data from the loop further away to perform RNC, thereby reducing the amount of SNMR-signal contamination in the remote reference loop (see discussion in Section 5.4.1).

In practice, a combination of different noise-filtering techniques is applied. We optimized the sequence of noise-reduction steps to maximize the S/N ratio and found the combination "RNC+DS+HNC+DS" to be the most effective for our case.
Note that this is different from the order most commonly found in the literature, i.e. "DS+HNC" and possibly RNC e.g. (Kremer et al., 2022; Müller-Petke et al., 2016; Larsen and Behroozmand, 2016). In Supplementary Fig. A1, we compare the noise remaining after different processing sequences and show that the combination "RNC+DS+HNC+DS" is actually the one leading to the best results. We discuss the remaining data uncertainty in Section 4.1 and the impact of the processing sequence on the model-parameter estimation in Section 5.4.



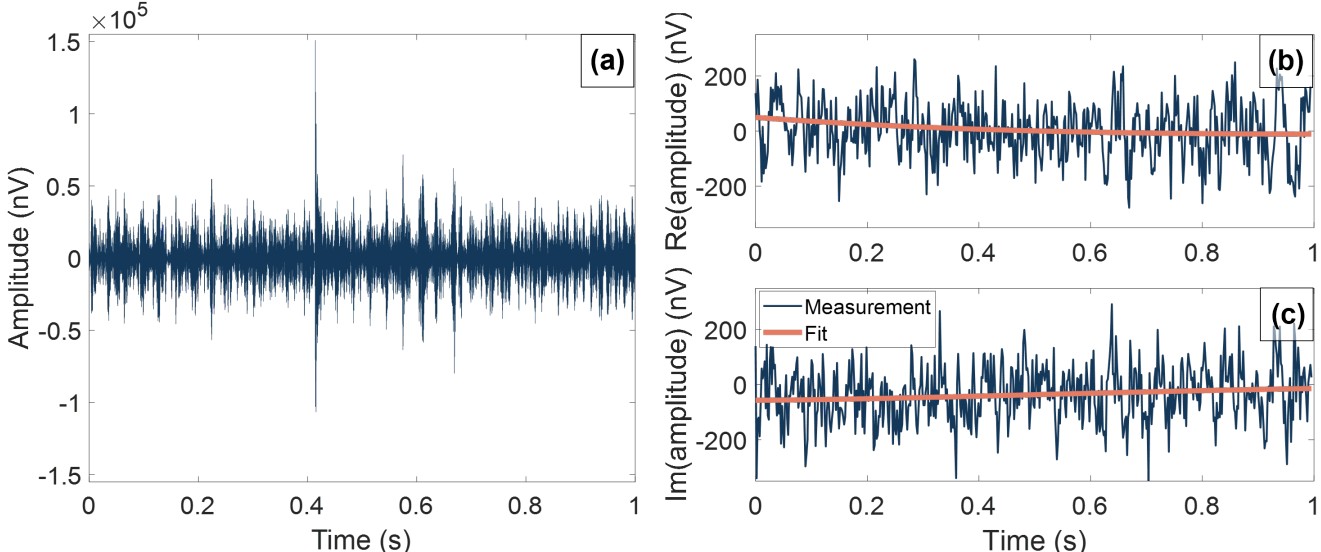

**Figure 3.** Exemplary raw and processed time series. (a) Raw signal time series recorded for one second. (b) Real and (c) imaginary parts of the processed times series, i.e. after noise reduction and envelope detection (blue). The orange line represents the fit based on the four estimated parameters $\mathbf{m} = (s_0, T_2^*, \delta f, \phi)$ (cf. Eq. 2).

**Table 1.** Overview of the selected measurement (left) and fitting parameters (right) of the SNMR survey.

| Measurement parameter | Value | Fitting parameter | Range |
|---|---|---|---|
| reference frequency $f_{\mathrm{r}}$ | 2039.2 Hz | amplitude $s_0$ | [0, 400] nV |
| pulse duration $\tau_{\mathrm{p}}$ | 40 ms | transverse relaxation time $T_2^*$ | [10, 1500] ms |
| pulse moments $q_i$ | 16 logarithmically-spaced values up to 8.8 As | frequency offset $\delta f$ | [-2, 2] Hz |
| recording time | 1 s | phase $\phi$ | [$-2\pi$ $2\pi$] rad |
| stacking number | 96 | | |

### 3.1.2 Envelope detection

Ultimately, only the envelope of the processed signal is relevant for the subsequent data interpretation (Müller-Petke et al., 2016). Again, we use the strategy implemented in MRSmatlab, as illustrated in the second column of Figure 2. First, the individual time traces are averaged (stacking). Next, the complex envelope is computed via a Hilbert transform and a low-pass filter (quadrature detection), and lastly, the time series are resampled. A more detailed description of the individual steps is found in Müller-Petke et al. (2016) while an exemplary complex envelope after noise reduction is presented in Figures 3b and 3c.





## 3.2 Model-parameter estimation

To identify water models based on the complex envelopes, we follow a two-step approach (Fig. 2, red part). First, we fit the processed time series to a mono-exponential decay, extracting so-called initial values, i.e. the initial amplitudes of the decay (for more information, see Sec. 3.2.1). Secondly, we perform a grid search in the model-parameter space to identify one-dimensional water models matching the previously found initial values. The grid search is conducted over a set of six different parameters (Fig. 4 for their definition and Sec. 3.2.2 for more information on the procedure), and is preferred over a deterministic inversion of the initial values (a so-called initial value inversion Mueller-Petke and Yaramanci, 2010; Legchenko and Shushakov, 1998), because of the poor S/N of our data set. Indeed, the latter makes an initial value inversion unfeasible. Note that more complex inversion techniques, such as QT-inversion (Mueller-Petke and Yaramanci, 2010), could provide information on both the spatial water and relaxation-time distributions. However, in our study, we focus solely on retrieving the water distribution as a function of depth, which justifies the use of the initial values approach. Furthermore, our method assumes a mono-exponential decay, meaning that spins contributing to a signal for a given pulse moment $q$ are assumed to exhibit similar relaxation times. We discuss this assumption and its implications in Section 5.4.2.

### 3.2.1 Mono-exponential fit

Assuming a mono-exponential decay, the complex envelope of the received SNMR signal can be expressed as a function of time (Müller-Petke et al., 2016):

$$s(q,t) = s_0(q)\mathrm{e}^{-\frac{t}{T_2^*(q)}}\mathrm{e}^{i(2\pi\delta f(q)t+\phi(q))} \tag{2}$$

where the four parameters $\mathbf{m}(q) = (s_0(q), T_2^*(q), \delta f(q), \phi(q))$ are directly related to subsurface properties:

- The amplitude $s_0(q)$ is a function of the distribution of water present in the subsurface. Based on $s_0$, the relaxation during the excitation pulse $\tau_\mathrm{p}$, and the dead time $\tau_\mathrm{d}$, we can retrieve the initial values $e_0(q)$ by extrapolating the amplitude to earlier times (Müller-Petke et al., 2011; Walbrecker et al., 2009):

$$e_0(q) = s\left(q, t = -(\frac{\tau_\mathrm{p}}{2} + \tau_\mathrm{d})\right) = s_0(q)\mathrm{e}^{\frac{\tau_\mathrm{p}/2+\tau_\mathrm{d}}{T_2^*(q)}}. \tag{3}$$

  The set of initial values $[e_0(q_1), e_0(q_2), ..., e_0(q_{15})]$ is thereby referred to as the sounding curve.

- The effective transverse relaxation time of the nuclear spins $T_2^*(q)$ depends on the material properties, like pore size, surface relaxivity of the surrounding solid material, temperature, or the concentration of paramagnetic species in the water (Behroozmand et al., 2015).

- The frequency offset $\delta f(q) = f_\mathrm{L}(q) - f_\mathrm{r}$ corresponds to the offset between the reference frequency $f_\mathrm{r}$ (set during acquisition, Tab. 1) and the local Larmor frequency $f_\mathrm{L}$. We expect a continuous variation of the frequency offset proportional to the changes in the geomagnetic field (Eq. 1).





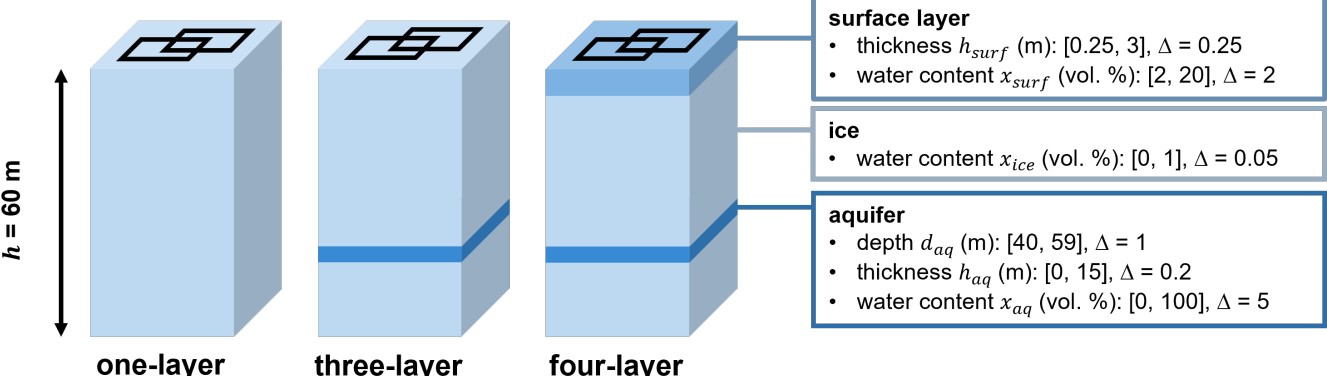

**Figure 4.** Schematic representation of the water models used to compute synthetic data according to Equation 5. The colours represent the different layers (surface, ice, or aquifer), which are parametrized according to the information in the boxes. The square brackets indicate the range and $\Delta$ the discretization used in the grid search. The squares on the surface of the columns schematically represent the transmitter and receiver loops. Note that the relative proportions of the layers are exaggerated for better visibility.

- The phase $\phi(q)$ can originate from off-resonance effects, variation in the electrical resistivity of the subsurface or internal effects of the instrument (e.g. Grombacher and Knight, 2015; Behroozmand et al., 2015).

We use the implemented fitting routines in MRSmatlab to estimate the parameters $\mathbf{m}(q)$ for each pulse moment. MRSmatlab searches for the maximum-likelihood model parameters using a least-squares approach within the range of values provided in Table 1. An example of the resulting fit is given in Figure 3b and 3c.

We assess the posterior uncertainties $\sigma_{\mathrm{m}}$ of the estimated parameters $\mathbf{m}(q)$ according to the covariance matrix $\tilde{C}_m \approx (G^T C_D^{-1} G)^{-1}$ at the maximum-likelihood point (Tarantola, 2005), where $C_D$ is the a priori data covariance, and $G$ is the linearized forward operator (Jacobian) of Equation 2. The data covariance is given by a diagonal matrix containing the data variances $\sigma_{\mathrm{D}}(q,t_i)^2$ retrieved from the ensemble of complex envelopes of the single recordings at each time sample $t_i$. As an approximation, the diagonal elements of $\tilde{C}_m$ correspond to the variance of the parameters $\mathbf{m}(q)$. In reality, the problem is nonlinear, and the parameters might not be normally distributed.

Ultimately, we are interested in the standard deviation of the initial value $e_0(q)$. Therefore, we need to estimate $\sigma_{e_0}$ from $\tilde{C}_m$. Assuming Gaussian error propagation, the uncertainty of $e_0(q)$ is then given as

$$\sigma_{e_0} \approx \mathrm{e}^{\frac{\tau_\mathrm{p}/2+\tau_\mathrm{d}}{T_2^*}} \sqrt{\sigma_{s_0}^2 + \sigma_{T_2^*}^2 s_0^2 \left( \frac{\tau_\mathrm{p}/2+\tau_\mathrm{d}}{T_2^{*2}} \right)^2}. \tag{4}$$



### 3.2.2 Forward problem and grid search

The initial value $e_0(q)$ obtained from the mono-exponential fit is related to the one-dimensional water distribution $f(z)$ according to

$$e_0(q) = \left| \int K(q,z) f(z) dz \right| \tag{5}$$

(Mueller-Petke and Yaramanci, 2010; Hertrich et al., 2005; Weichman et al., 2000; Legchenko and Shushakov, 1998), where $K(q,z)$ corresponds to the kernel as a function of depth $z$ and pulse moment $q$. The kernel relates the response of the subsurface to a magnetic perturbation (emitted by the transmitter loop) with the resulting measurable voltage in the receiver loop. Consequently, $K(q,z)$ depends on the loop configuration, the measurement parameters (Tab. 1) and the material properties of the subsurface. In this study, we compute the kernel for both the coincident- and separate-loop configurations (cf. 1b) using the

functionalities of MRSmatlab. We simplify the computation by assuming a highly resistive subsurface – a reasonable approximation for glacier ice (Kulessa, 2007). To further simplify the computation of the magnetic fields, we use circular loops with the same area as the square loops, which should result in a minor difference for large loops (Kremer et al., 2019).

Based on the kernel $K(q,z)$ and the initial values $e_0(q)$, we aim to infer possible water-content distributions $f(z)$. Given

that previous measurements (Church et al., 2021) let us expect a broad yet thin conduit embedded in ice, we select water-model parametrizations that include different layers representing the channel and the glacier ice. More specifically, we consider three simplified models (Fig. 4): The one-layer model consists of a 60 m thick, uniform ice column with a homogeneous liquid-water content (LWC) $x_{ice}$ (1 parameter), where the ice thickness reflects the information from our GPR data (cf. Sec. 5.2). The three-layer model consists of the same ice column but additionally includes an aquifer of thickness $h_{aq}$ at depth $d_{aq}$ ($d_{aq}$ being

defined as the upper boundary of the layer) with LWC $x_{aq}$ (4 parameters in total when including $x_{ice}$). This layer is meant to represent the englacial water channel. Finally, the four-layer model builds on the three-layer model but includes a separate surface layer of thickness $h_{surf}$ and LWC $x_{surf}$ (6 parameters in total). This surface layer is meant to present a weathering ice crust as is typically found on glacier surfaces (e.g. Müller and Keeler, 1969).

The combination of water-model parameters are sufficient to define the water-content distribution of the four-layer model as

a function of depth $z \in [0, 60]$ m:

$$f(z) = \begin{cases} x_{surf}, & \text{if } 0 \leq z < h_{surf} \\ x_{aq}, & \text{if } d_{aq} \leq z < d_{aq} + h_{aq} \\ x_{ice,} & \text{otherwise} \end{cases} \tag{6}$$

. For the one- and three-layer models, instead, it holds that $x_{surf} = x_{aq} = x_{ice}$ (one layer) and $x_{surf} = x_{ice}$ (three layers).

We perform a grid search within the parameter space spanned by $(x_{ice}, h_{aq}, d_{aq}, x_{aq}, h_{surf}, x_{surf})$ to identify the most likely water distributions $f(z)$ explaining the measured $e_0(q)$. For all possible combinations of $(x_{ice}, h_{aq}, d_{aq}, x_{aq}, h_{surf}, x_{surf})$, we

repeat the following three steps (cf. Fig. 2):





1. **Computation of synthetic data**: Based on the kernel $K(q, z)$ for a given loop configuration and a set of water-model parameters $(x_{ice}, h_{aq}, d_{aq}, x_{aq}, h_{surf}, x_{surf})$, we compute the synthetic sounding curve $e_0^{syn}(q_i)$ for the set of pulse moments $q = [q_1, q_2, ..., q_{15}]$ according to the forward problem in Eq. 5.

2. **Computation of $\chi^{\text{RMS}}$**: To compare the synthetic sounding curve $e_0^{syn}(q_i)$ to the measured one $e_0(q_i)$, we compute the
error-weighted root-mean-square (RMS) misfit $\chi^{\text{RMS}}$ according to (Fichtner, 2021)

$$\chi^{\text{RMS}} = \sqrt{\frac{1}{N} \sum_{i=1}^{N} \frac{\left((e_0(q_i) - e_0^{syn}(q_i))\right)^2}{\sigma_{e_0(q_i)}^2}} \tag{7}$$

, where $N = 15$ is the number of pulse moments.

3. **Selection of compatible models**: Any water model described by $(x_{ice}, h_{aq}, d_{aq}, x_{aq}, h_{surf}, x_{surf})$ resulting in $\chi^{\text{RMS}}$ below
    a threshold value $\chi_{\max}^{\text{RMS}}$, is retained and sorted according to its $\chi^{\text{RMS}}$. We set the threshold value to $\chi_{\max}^{\text{RMS}} = 1.9$, which is
a compromise between computational effort and the number of models retained for analysis.

We perform the grid search above for two data sets $e_0(q_i)$: The first data set entails only the coincident-loop data, i.e. $e_0(q_i) = e_{0,\text{coi}}(q_i)$. The second data set combines both the coincident- and separate-loop data, resulting in a joint set of initial values $e_{0,j}(q_i)$, where $j = \text{coi, sep}$. We select the best parameters based on the joint $\chi_{\text{joint}}^{\text{RMS}}$ given as

$$\chi_{\text{joint}}^{\text{RMS}} = \sqrt{\frac{1}{2N} \sum_{j=\text{coi, sep}} \sum_{i=1}^{N} \frac{\left((e_{0,j}(q_i) - e_{0,j}^{syn}(q_i))\right)^2}{\sigma_{e_{0,j}(q_i)}^2}}. \tag{8}$$

## 4    Results

### 4.1    Data interpretability

After processing the data according to the scheme in Figure 2, the signal-to-noise ratio of the time series increased significantly. While the application of DS and HNC slightly improved the S/N, the application of RNC was essential to reduce the noise level by an order of magnitude (Fig. A1). We compare the average noise before and after processing by calculating the average
data uncertainty $\overline{\sigma}_D$ (Müller-Petke et al., 2011). Assuming that the standard deviations $\sigma_D(q, t_i)$ are independent, we take the average over all time samples, pulse moments and the real and imaginary parts of the time series, and retrieve an estimation of the mean data uncertainty of the complete data set $\overline{\sigma}_D$. The complex envelopes of the raw traces (i.e. without noise reduction) of the coincident-loop measurement show an average data uncertainty of $\overline{\sigma}_D \approx 1600 \, \text{nV}$. After processing, including noise reduction, the remaining uncertainty amounts to $\overline{\sigma}_D \approx 70 \, \text{nV}$ – a reduction in noise by a factor of $\approx 23$. However, despite this
improvement, the S/N remains poor, and the mono-exponential decay is not evident in the time traces (Figs. 3b+c). Nonetheless, we can confirm the existence of an SNMR signal by studying the complete, processed data set in the frequency domain (Fig. 5).

Figure 5a illustrates the spectral content of the time traces recorded after the excitation pulse, thus containing both noise and SNMR signal (signal traces). In contrast, Figure 5b shows the time series recorded before the excitation pulse (noise traces),



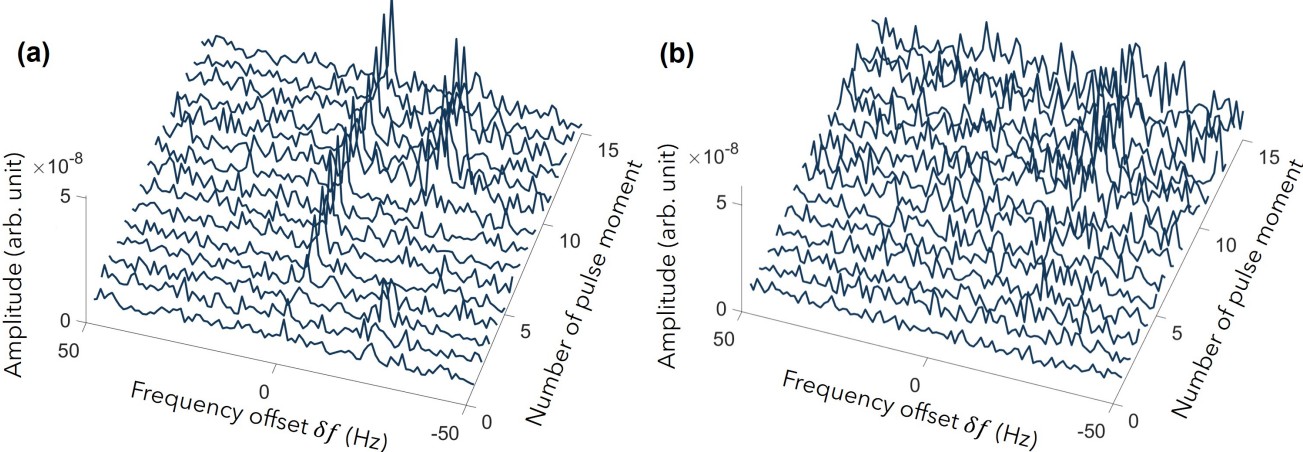

**Figure 5.** Representation of the processed data set in the frequency domain (after a fast Fourier transform of the time series) as a function of pulse-moment number. Note that the frequency is given in terms of the frequency offset $\delta f$ (cf. Sec. 3.2.1). (a) Spectra of the signal traces (measured after the excitation pulse). (b) Spectra of the noise-only traces (measured before the excitation pulse).

not recording any SNMR signal. The noise traces do not show any increased amplitude close to the expected Larmor frequency, i.e. where $\delta f = 0$ Hz. In comparison, most of the signal traces exhibit higher amplitudes centred around 0 Hz, clearly showing the presence of an SNMR signal. The peaks at around -20 Hz indicate the presence of some residual higher harmonics that could not be removed with our processing routine.

Next, we assess the results of the mono-exponential fit, which provides the basis for the subsequent water-model estimation. Figure 6 presents the estimated parameters and their standard deviations for both the coincident-loop (blue) and separate-loop (orange) measurements. In general, the uncertainties of the parameters vary between pulse moments, reflecting the variability of the noise and the quality of the fits. Figure 6a depicts the estimated initial values $e_0(q_i)$ as a function of the pulse moments $q_i$ (sounding curve). We observe amplitudes between 0 and 110 nV corresponding to roughly the order of magnitude of the average noise level (70 nV) for most pulse moments. Figure 6b presents the corresponding relaxation times $T_2^*(q_i)$. They range between 150 ms and 1500 ms some having significant uncertainties. The upper bound of the observed range corresponds to the maximum possible relaxation time allowed in the fit. Fits yielding $T_2^* \approx 1500$ ms generally come with larger uncertainties and correspond to low initial values ($e_0 < 30$ nV in Fig. 6a). We attribute this finding to the poorer fits associated with lower S/N. Values of $T_2^*$ closer to the lower bound of the observed range generally correlate with higher initial values $e_0$. A negative correlation between initial value and relaxation time estimations has been shown before (Mueller-Petke and Yaramanci, 2010) and is consistent with our observations. This means that a misestimation of one of the parameters may be compensated by a misestimation of the other parameter.

Figure 6c displays the estimated frequency offset. The offset varies continuously and displays a similar trend for both configurations, except at lower pulse moments. According to Equation 2, the frequency of the mono-exponential decay is linearly related





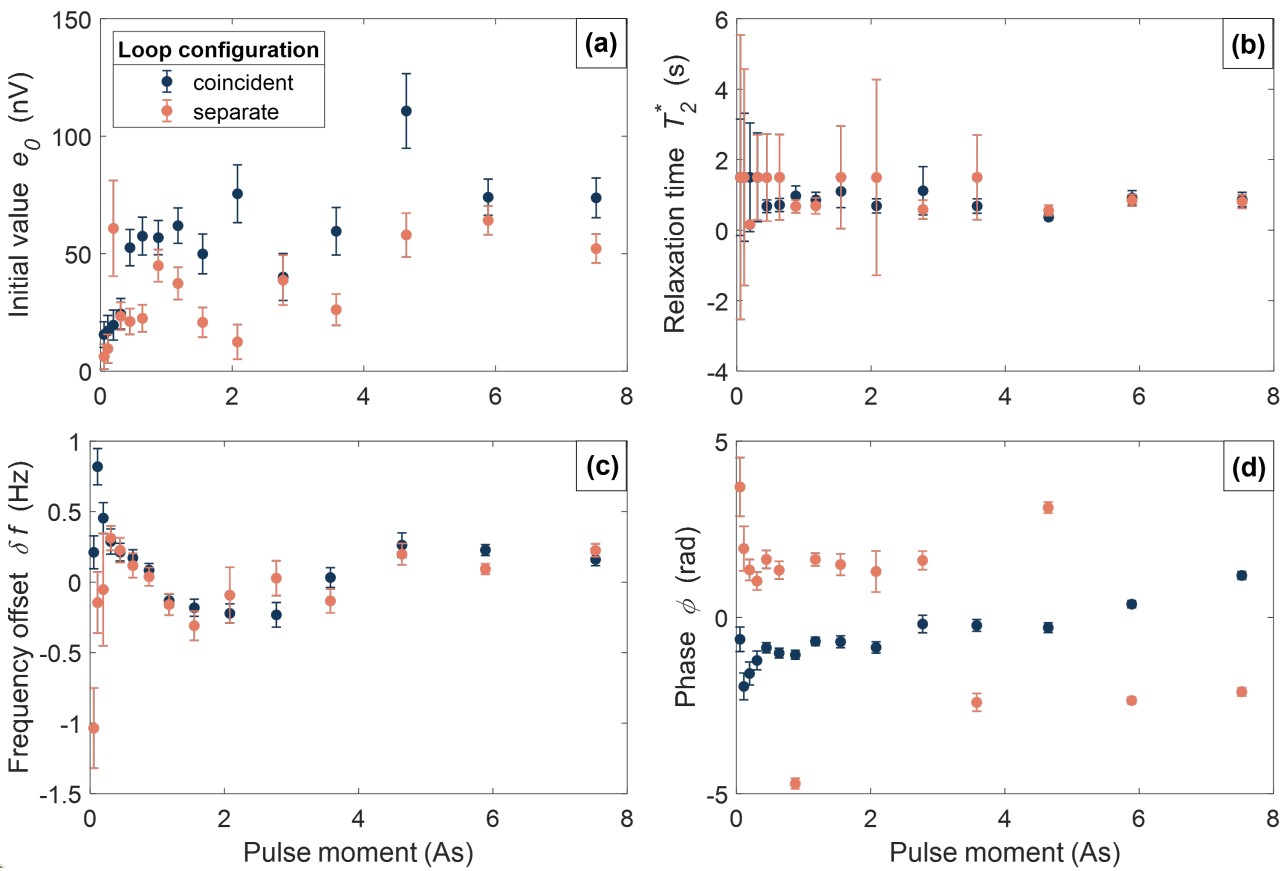

**Figure 6.** Estimation of the parameters from the mono-exponential fit (cf. Eqs. 2 and 3) with corresponding uncertainties as a function of pulse moment. The coincident-loop data and the separate-loop data are shown in blue and orange, respectively. (a) Initial value $e_0$, (b) relaxation time $T_2^*$, (c) frequency offset $\delta f$, and (d) phase $\phi$.

to variations in the geomagnetic field, which naturally occur during the day (generally over a few nT). In the Supplementary Figure A2, we plot the correlation between the frequency offsets $\delta f$ (cf. Fig. 6c) and the geomagnetic field amplitude recorded simultaneously at the Black Forest Observatory (Intermagnet). The Pearson correlation amounts to 0.78 for the coincident-loop measurements and to either -0.21 (when considering all data points) or 0.67 (after excluding the three data points taken at the lowest pulse moments, exhibiting the poorest S/N) for the separate-loop measurements. The high correlation for the latter configuration indicates that the obtained parameters are indeed derived from fitting a real SNMR signal, rather than just noise.

Theoretically, we expect $\phi = 0$ for the coincident measurement assuming a resistive subsurface (Hertrich, 2008). However, we observe a phase $\phi \neq 0$, which likely originates from variable instrumental phases, off-resonance effects, or processing. The separate-loop configuration exhibits a phase shift of $\pm\pi$ compared to the coincident-loop measurements, which stems from an opposite polarity of the loop. We do not further interpret the phase since we cannot identify the exact origin of it.



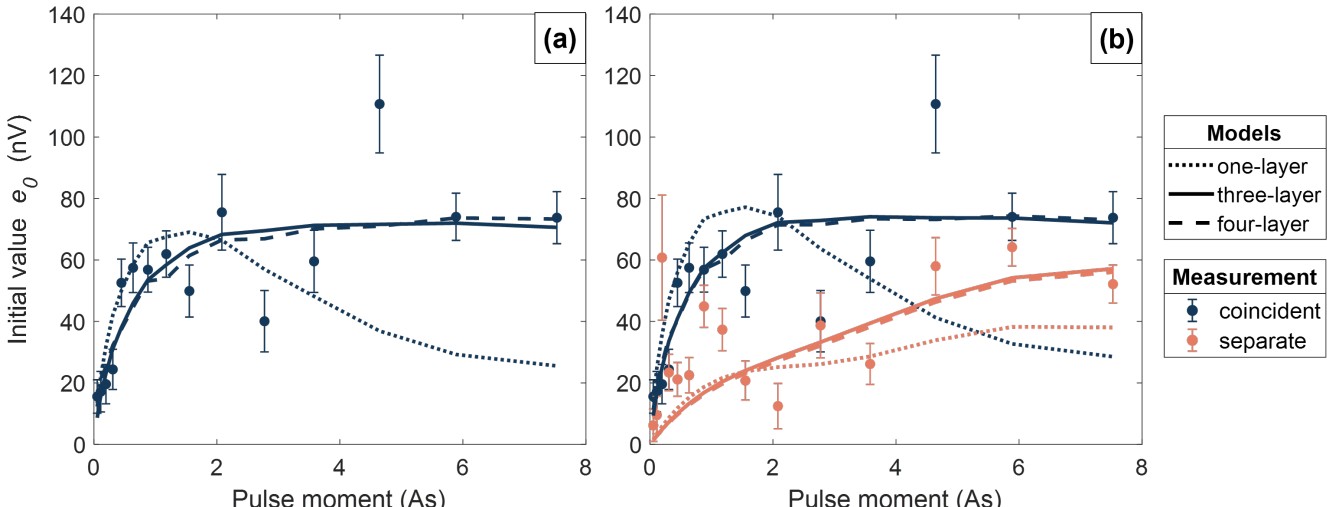

**Figure 7.** Comparison of the measurements (dots with error bars) and the synthetic sounding curves based on the minimum RMS-misfit models (lines) for the coincident-loop (blue) and the separate-loop configuration (orange). The different line types correspond to the three different models presented in Figure 4. (a) Comparison of the synthetic and measured sounding curve based on the coincident-loop configuration. (b) Comparison of the synthetic and measured sounding curve based on the coincident- and separate-loop configuration (joint data).

**Table 2.** Summary of the parameter ranges for $\chi^{\text{RMS}} \leq 1.9$ and parameter values leading to the minimum RMS-misfit (denoted "at min. $\chi^{\text{RMS}}$" in the table) for both the coincident-loop data (left column) and the combination of coincident- and separate-loop data (right column). Parameter ranges with multiple minima show the entry "-" for minimum RMS-misfit.

| Parameter | coincident | | coincident and separate | |
|---|---|---|---|---|
| | range | at min. $\chi^{\text{RMS}}$ | range | at min. $\chi^{\text{RMS}}$ |
| ice water content (vol. %) | 0.30 - 0.75 | 0.55 | 0.40 - 0.75 | 0.60 |
| aquifer depth (m) | 41 - 59 | 59 | 43 - 59 | 59 |
| aquifer thickness (m) | 0.2 - 13.6 | - | 0.4 - 12.4 | - |
| aquifer water content (vol. %) | 5 - 100 | - | 5 - 100 | - |
| aquifer water volume (m$^3$/loop area) | 1800 - 7200 | 4800 | 2400 - 6500 | 4800 |
| total water volume (m$^3$/loop area) | 5370 - 9952 | 8056 | 6270 - 9564 | 8352 |

In conclusion, we confirm the existence of an SNMR signal in the raw data, which we could fit with a mono-exponential decay extracting the initial values necessary for the model-parameter estimation.





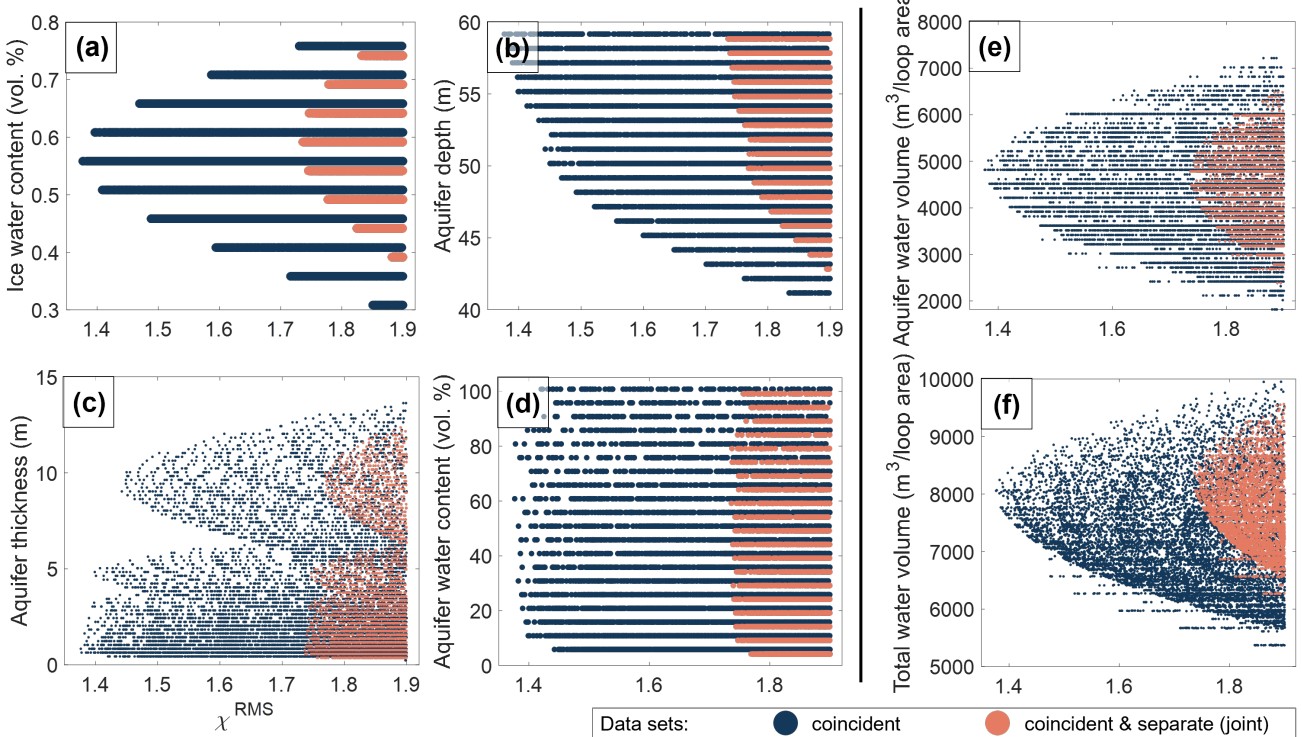

**Figure 8.** Compatible parameter ranges for $\chi^{\mathrm{RMS}} \leq \chi^{\mathrm{RMS}}_{\max} = 1.9$ for the three-layer models based on the grid search. The x-axis corresponds to $\chi^{\mathrm{RMS}}$ (cf. Eqs. 7 and 8), while the y-axis represents either the (a) water content of the ice, (b) aquifer depth, (c) aquifer thickness, or (d) water content of the aquifer. The parameters aquifer water volume (e) and total water volume (f) are derived from the parameters in (a) - (d) (cf. Sec. 4.2.2). Each dot corresponds to a combination of model parameters with $\chi^{\mathrm{RMS}} \leq \chi^{\mathrm{RMS}}_{\max}$, the color of the dot discerning between coincident-loop measurements (blue) and the combination of the coincident- and separate-loop measurements (orange).

## 4.2 Compatible water models

### 4.2.1 Minimum RMS-misfit models

Figure 7a presents the synthetic sounding curves with the smallest $\chi^{\mathrm{RMS}}$ obtained for each class of models (i.e. one-, three- or four-layer models) for the coincident measurement. With $\chi^{\mathrm{RMS}} = 2.73$, the one-layer model exhibits the lowest agreement with the observations. Specifically, the model fails to reproduce the amplitudes at higher pulse moments. The best-performing three-layer model shows significantly better agreement with the observations ($\chi^{\mathrm{RMS}} = 1.38$) while adding an additional surface layer, as done in the four-layer model, yields only minimal further improvement ($\chi^{\mathrm{RMS}} = 1.37$).

We conduct the same analysis for the joint set of initial values consisting of the coincident- and separate-loop data (Fig. 7b). The best performing synthetic sounding curves yield a $\chi^{\mathrm{RMS}}_{\mathrm{joint}}$ of 2.56, 1.74 and 1.75 for the one-layer, three-layer, and four-layer models, respectively. We thus observe the same trends as in the previous case: Even the best one-layer model shows a poor fit,





while the best three- and four-layer models result in very similar synthetic sounding curves and, thus, $\chi_{\mathrm{joint}}^{\mathrm{RMS}}$. Based on these observations, we only consider three-layer models from now on.

If the synthetic data $e_0(q_i)$ fit all of the observations within their observational uncertainty $\sigma_{e_0(q_i)}$, we expect $\chi^{\mathrm{RMS}} \approx 1$. In our case, none of the models reaches this value, suggesting a slight under-fitting. This under-fitting could be an expression of

our simplified forward problem (cf. Sec. 5.4.2) or of a misestimation of the initial value's uncertainty. For instance, even the best model fails to replicate the amplitudes at lower pulse moments for the separate-loop data (Fig. 7b).

### 4.2.2 Ensemble of low RMS-misfit models

To investigate the relationship between parameter ranges and RMS misfit, we consider an ensemble of three-layer models resulting in an RMS misfit below a certain threshold. Figure 8 and Table 2 present the range of compatible model parameters

for $\chi^{\mathrm{RMS}} \leq 1.9$. This threshold is arbitrary to a large degree, but the intention is to retain a sufficient number of models for both the coincident-loop data (orange dots in Fig. 8) and the joint data (blue dots). Increasing the threshold would result in broader parameter ranges but also result in the selection of models that are less likely.

The ice water content (Fig. 8a) and the aquifer depth (Fig. 8b) show a continuous broadening of the parameter range as $\chi^{\mathrm{RMS}}$ increases. The distribution of the corresponding parameter values is parabola-like, with a vertex corresponding to the

minimum $\chi^{\mathrm{RMS}}$. The aquifer thickness (Fig. 8c) is anti-correlated with the aquifer water content (Fig. 8d), meaning that in terms of $\chi^{\mathrm{RMS}}$, situations with a thick but water-poor aquifer are virtually indistinguishable from situations in which the aquifer is thin but water-rich (Supplementary Figure A3). Therefore, establishing a minimum RMS misfit for the aquifer water content alone is not particularly informative.

The above considerations let us introduce two additional parameters: (1) The aquifer water volume $V_{\mathrm{aq}}$ normalized by the loop

area (Fig. 8e), corresponding to the product of the aquifer water content and the aquifer thickness (i.e. $V_{\mathrm{aq}} = x_{\mathrm{aq}} \cdot h_{\mathrm{aq}}$), and (2) the total water volume $V_{\mathrm{water}}$ normalized by the loop area (Fig. 8f), corresponding to the sum of the aquifer water volume and the product of the ice water content and its total thickness (i.e. $V_{\mathrm{water}} = V_{\mathrm{aq}} + x_{\mathrm{ice}} \cdot (h - h_{\mathrm{aq}})$, where $h$ is the total thickness of our three-layer model). Both parameter ranges exhibit a parabola-like distribution.

In general, the coincident data show lower $\chi^{\mathrm{RMS}}$ than the joint data (Fig. 8) although the two distributions follow similar

patterns and result in similarly low minimum $\chi^{\mathrm{RMS}}$ values (Tab. 2). We explain the small differences with the one-dimensional nature of our simplified subsurface: If the subsurface was perfectly one-dimensional, the models should fit the coincident- and separate-loop data equally well. In reality, the subsurface likely exhibits a three-dimensional water distribution (cf. Sec. 5.4.2) and the joint dataset contains more information about the water distribution compared to the coincident dataset.




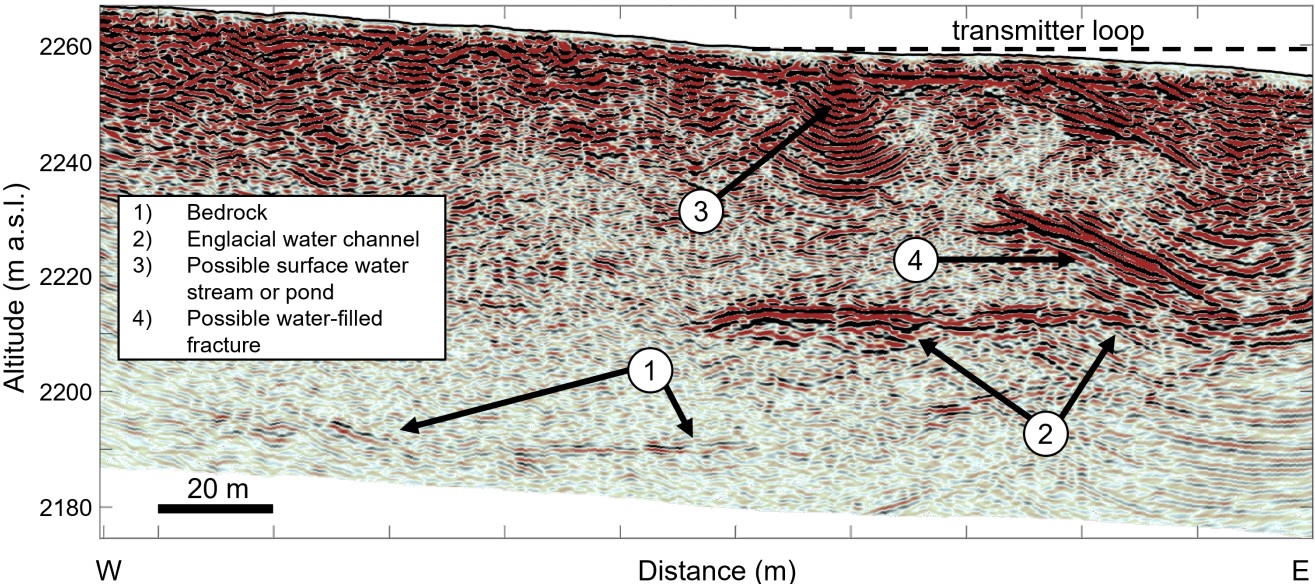

**Figure 9.** GPR profile (50 MHz) acquired in the SNMR survey area (Fig. 1 for location) showing different features of interest. The interpretation of the different features is indicated by the legend. The dashed line in the right corner represents the approximate extent of the transmitter loop along the W-E direction (x-axis).

## 5 Discussion

### 5.1 Plausibility of the most likely water models

In this section, we interpret the parameter ranges of the most likely three-layer models (Fig. 8) from a glaciological perspective and compare these values with values reported in the literature.

The findings indicate an ice liquid water content (LWC) between 0.30% and 0.75% (coincident-loop data). The model with the smallest $\chi^{\mathrm{RMS}}$ indicates LWC = 0.55%. This is towards the lower end but well within the range of values reported in the literature. Pettersson et al. (2004), for example, reviewed various studies investigating volumetric LWC in temperate and polythermal glaciers worldwide. They report values between 0 and 9%, typically based on calorimetric measurements, GPR measurements, or a combination of both. When only selecting studies performed on Alpine glaciers (all based on calorimetric measurements), the range is narrowed to between 0 and 3% (Pettersson et al., 2004).

The estimated aquifer thickness ranges between 0.2 and 13.6 meters, with corresponding LWC values between 5% and 100%. These broad ranges can be explained by the strong correlation between the two parameters (Supplementary Fig. A3), which makes it impossible to discern between combinations resulting in similar synthetic data. However, we can still identify models that are more likely than others, based on prior knowledge of the channel system: Previous GPR surveys (Church et al., 2021, 2020) estimated a conduit's thickness below 0.4 m, which is at the lower limit of the thicknesses suggested by



our study. In addition, from a glaciological standpoint, we expect a water content close to 100 % given that the conduit is
likely primarily filled with water (and possibly some air). We can thus assume that models with thin aquifers and high LWC
are closest to reality. Moreover, assuming a one-dimensional aquifer instead of a three-dimensional channel, likely results in
an underestimation of the liquid water content in the aquifer. Based on these considerations, we show parameter ranges with
$x_{aq} \geq 60\%$ and $\chi^{RMS} \leq 1.9$ in Supplementary Figure A4. By doing so, the range of aquifer thicknesses decreases drastically,
allowing for values between 0.2 and 1.0 meters. The ranges for the other parameters remain very similar to the ones in Figure 8.

The aquifer depth of the compatible models varies between 41 and 59 meters, with the minimum $\chi^{RMS}$ corresponding to
the deepest aquifers (59 m). We further discuss the depth in the next section, comparing it to data obtained from the GPR
measurements performed in the same area.

## 5.2  Validation with GPR data

Our glaciological interpretation is corroborated by the GPR survey conducted in the study area. Figure 9 shows an exemplary
GPR profile acquired in the area of the transmitter loop. A distinct, horizontal reflection is visible at around 2214 meters above
sea level, and this feature is consistently observed in other GPR profiles collected in the area too (not shown). We interpret
these reflections as the englacial channel, as they appear at locations that are consistent with the water channel identified in
earlier studies (Church et al., 2021).

From the GPR data, the average depth of the channel is around 40 m below the transmitter loop (Fig. 9). This is somewhat
shallower, but broadly consistent with the parameter distributions obtained from our SNMR investigations, which indicate a
channel depth between 41 and 59 meters (Fig. 8b).

In addition to the channel, the GPR signals also reveal weak bedrock reflections and various features that we interpret as
being part of the glacier's drainage system (including a surface water streams and possibly, a water-filled fracture; cf. Fig. 9).
The spatial distribution of these latter, partially englacial features, indicate that our one-dimensional water models (cf. Fig. 4)
might be an oversimplification as all of them have variable, three-dimensional shapes. In Section 5.4, we further discuss this
limitation and how it might help explain the discrepancy between the channel depth as estimated from SNMR and GPR.

## 5.3  High SNMR noise levels and possible origins

During the SNMR survey on the Rhonegletscher in August 2023, the coincident-loop measurements showed an average noise
level of $\approx 1.6\,\mathrm{nV\,m^{-2}}$ (average over the standard deviations of the single raw time series recorded before the excitation pulse).
This value is relatively high compared to those reported in the literature (Müller-Petke, 2020; Larsen and Behroozmand, 2016;
Lehmann-Horn et al., 2011; Legchenko et al., 2011). For example, Larsen and Behroozmand (2016) studied the noise properties
of multiple sites in Denmark. They investigated "sites with high-noise levels" showing an RMS misfit of 0.25 and $0.3\,\mathrm{nV\,m^{-2}}$,
which is almost one order of magnitude lower than the noise we recorded.

Considering the location of the two studies, this difference is remarkable: The site in Denmark is located in a village near
Aarhus, and high noise can thus be expected due to the proximity to electrical infrastructure. Rhonegletscher, on the contrary,
is located in a relatively remote area of the Swiss Alps with no evident source of electromagnetic noise. To our knowledge, the





closest potential sources are a hydropower plant (located at $> 4\,\text{km}$ distance), a road (at $\approx 1\,\text{km}$), a railway tunnel (at $> 1.5\,\text{km}$) and some military infrastructure (at $\approx 2\,\text{km}$). Since no thunderstorms were recorded in the larger area during the survey either, we remain puzzled by the noise's origin. While the data exhibits some signatures of spikes and higher harmonics of 16.6 and 50 Hz, the predominant noise is probably a superposition of multiple sources.

Our noise levels are also one or two orders of magnitude higher than that reported for an SNMR study conducted on Tête Rousse Glacier, France (Vincent et al., 2012; Legchenko et al., 2011). In that case, the SNMR campaign was performed in the summer of 2009, and noise levels ranged between 0.03 and 0.125 $\text{nV}\,\text{m}^{-2}$.

We claim that further research is necessary to better understand the spatial and temporal characteristics of electromagnetic noise in Alpine environments, and hypothesize that the dense infrastructure in the Swiss Alps might be the cause of the substantial electromagnetic noise we encountered. Since the high noise levels have implications for both data processing and model-parameter estimation (see next section), the topic is relevant for future SNMR studies.

### 5.4 Limitations of the workflow

#### 5.4.1 Impact of processing on SNMR signal estimation

RNC is the most crucial step in our noise cancellation sequence. For optimal noise cancellation, one wants to maximize the correlation between the time series of the remote reference loops and the receiver loop while detecting the SNMR signal exclusively in the receiver loop. If the noise is strongly correlated over large distances, the remote loop could be placed sufficiently far away from the transmitter loop, thereby avoiding SNMR-signal contamination. In our case, we deployed a remote reference loop at a distance of $\approx 120\,\text{m}$ (center-to-center), which makes SNMR-signal contamination likely (Kremer et al., 2022). The distance was constrained by both the terrain and cable length, and chosen to cope with the heterogeneous nature of the noise.

Based on the minimum RMS-misfit model found in our analysis, we would expect an SNMR-signal amplitude of up to $\approx$ 1.9 nV for the highest pulse moment in the remote reference loop (cf. Supplementary Fig. A5). We base this estimation on the synthetic sounding curve computed according to Eq. 5 assuming a separate-loop configuration with a center-to-center distance between Tx and RNC loops of $\approx 122\,\text{m}$. To get a preliminary estimate of the maximum amplitude of the signal distortion in the receiver loop, we multiply the estimated signal amplitude ($\approx 1.9\,\text{nV}$) with the ratio of the effective areas of the receiver loop ($100 \cdot 100 = 10{,}000\,\text{m}^2$) and the reference loop ($7 \cdot 10 \cdot 10\,\text{m}^2 = 700\,\text{m}^2$), thereby including the effect of the loop size and number of turns. Based on this simple calculation, we expect distortions of up to 27 nV for the highest pulse moment, which is significant and thus introduces an additional uncertainty. Note that the distortion can be positive or negative depending on the phase imposed by the transfer function. A misestimation of this magnitude would also affect the estimation of the most likely water models.

To mitigate this uncertainty in future studies, one could characterize the noise field in advance, and optimize the positioning of the remote reference loop in order to balance the SNMR-signal contamination with noise correlation. Alternatively, one could include the SNMR-signal contamination in the inversion method, performing so-called non-remote reference noise cancellation



(Müller-Petke, 2020). However, for the non-remote approach, precise knowledge of the position and geometry of the reference loops is necessary.

While RNC is likely to result in the largest signal distortion, the specific sequence of the various processing steps also influences the resulting SNMR signal. For example, the sequence "RNC+DS+HNC+DS" may yield slightly different results than "DS+HNC+RNC+DS", as shown in Supplementary Fig. A6. Therefore, one should be aware that in the case of low S/N,
variations in processing sequences can influence the estimated water model parameters.

### 5.4.2 Limitations of the model-parameter estimation

**Mono-exponential decay:** Our parameter estimation for the water model is based on a mono-exponential decay (Eq. 2). This assumes that spins contributing to a signal for a given pulse moment $q$ exhibit similar relaxation times. From our mono-exponential fit, we find values of $T_2^*$ between 150 and 1000 ms (Fig. 6). Here, we discuss (1) the potential for multi-exponential
decay instead of mono-exponential decay, and (2) the expected range of relaxation times in glacier ice.

Multi-exponential decays are typically observed in media with a broad pore-size distribution and high surface relaxivities (Behroozmand et al., 2015). In glaciers, we expect the water to be contained in structures ranging from $\mu$m, such as veins or lenses between ice grains (Fowler and Iverson, 2023; Fountain and Walder, 1998; Lliboutry, 1996; Raymond and Harrison, 1975), to several meters, such as the englacial channel (Church et al., 2021). Beyond the pore size, the relaxation time strongly
depends on the local chemical composition of the pores, including the surface relaxivity of interfaces and the concentration of paramagnetic impurities in the liquid (Behroozmand et al., 2015). Impurities can, for example,accumulate in the water present between ice grains (Cuffey and Paterson, 2010), which could affect the relaxation time. The concentration of impurities might also vary across the glacier, as shown by Brown (2002); Brown and Fuge (1998) who investigated the concentrations of impurities in the meltwater of Haut Glacier d'Arolla, Switzerland. The concentrations of ions and trace elements in supraglacial
water were lower than in meltwater that had already passed the glacial drainage system. Based on these considerations, we cannot dismiss the possibility that the data collected from Rhonegletscher may, in fact, reflect a multi-exponential decay. However, with the current S/N, resolving multiple decays is out of range.

The longest relaxation times are expected for the water in the channel, with values expected to be close to the ones found for larger water bodies (up to 1.5 s Grunewald and Knight, 2011; Schirov et al., 1991). In contrast, water present between ice
grains likely exhibits the shortest relaxation times. Due to the complex interplay between impurity concentration, pore size, pressure, temperature and liquid water content (see e.g. Lei et al., 2022, for a study on the liquid vein network in frozen brine), an estimation of $T_2^*$ is not possible, and further research is necessary in this area.

**Forward model:** Due to the poor S/N of our dataset and based on prior knowledge of englacial hydrology, we opted for a low-dimensional parametrization of the considered water model space. Previous studies (Church et al., 2021, 2020, 2019) and
the GPR survey conducted in 2023 (Fig. 9) indicate that the englacial drainage system and the surrounding ice and bedrock, exhibit a three-dimensional structure. Consequently, our one-dimensional water models are a significant simplification of reality. We argue that for the given S/N, a higher-dimensional parameter space would not yield improvements, as the additional pa-



rameters would be poorly constrained. Thus, we deemed a full three-dimensional subsurface tomography out of scope. Instead, by performing a grid search, we identify and analyze the most likely three-layer models according to $\chi^{RMS}$.

With $\chi^{RMS} = 1.37$ being the minimum misfit, we are confident that the selected models provide insights into possible water distributions. For instance, our findings suggest that a three-layer structure, including a deep aquifer is much more likely than a structure with one layer only. Although we cannot resolve the exact depth and thickness of the aquifer, we can provide information on the range of possible parameters, and the degree to which the resulting synthetic data align with our observations. In future studies, a higher resolution could be achieved by either increasing the S/N or by performing multiple soundings in
different locations (Hertrich et al., 2009). The former could be achieved with higher stacking numbers, better noise cancellation techniques or by selecting time windows where the noise is the lowest.

## 5.5   Potential relevance of our findings

The total water volume within a glacier, which is a quantity we can resolve (Fig. 8e+f), can be an important indicator for the assessment of natural hazards. For example, Vincent et al. (2012) performed an extensive SNMR study combined with GPR
measurements on a water-filled reservoir within Tête Rousse Glacier, French Alps. They estimated a total water volume of $55,000\,\text{m}^3$, which posed a hazard for the downstream valley in case of an outburst. Based on their studies, most of the water was artificially pumped out of the reservoir, effectively mitigating the hazard (Vincent et al., 2015).

Based on the model with minimum $\chi^{RMS}$, our study estimates a total water volume of about $8,000\,\text{m}^3$ under the loop area of $10,000\,\text{m}^2$. Due to the relatively small volume of water and the continuous drainage of most of it through the channel, we
do not anticipate any actual risk in the case of Rhonegletscher. Nevertheless, our approach can be used to estimate the water volume present sub- or englacially in a selected area of a glacier. In particular, we demonstrate that a single survey is sufficient to provide an order-of-magnitude estimate of the corresponding water volume in the survey area. This could be helpful for future investigations assessing the risk of englacial outburst floods, linked e.g. to the rupture of englacial water pockets (Ogier et al.).

# 6   Conclusions

In this proof-of-concept study, we demonstrated that despite high background noise levels, it was possible to use SNMR to detect an englacial channel on Rhonegletscher, Swiss Alps. In terms of channel location and size, our findings are broadly consistent with GPR data acquired both in the frame of this work and in earlier studies (Church et al., 2021).

Despite exceptionally high noise levels, we successfully detected SNMR signals by carefully optimizing the data-processing
workflow. We identified remote reference noise cancellation as the most crucial step to increase the S/N. Based on the initial values extracted from a mono-exponential fit, we performed a grid search to identify water models compatible with our SNMR data. The most likely models consist of an ice column intersected by a deep aquifer representing the channel. Assuming a minimum aquifer water content of $60\,\%$, the selected models (with $\chi^{RMS} \leq 1.9$) indicate a thin ($< 1$ m) layer close to the bed





(44 - 60 m depth), embedded in an ice column with a LWC between 0.3 and 0.75 % (cf. Supplementary Fig. A4). Albeit low,

this LWC is compatible with values found in the literature.

We carefully examined the limitations of the data-processing workflow and model-parameter estimation procedure. Our results indicate that applying RNC can lead to significant signal distortions, which may impact subsequent water model estimates. We also show that the sequence of processing steps might influence the parameter estimates from the mono-exponential fit, especially under low signal-to-noise conditions. For our parameter estimation, we relied on a mono-exponential decay model,

thereby neglecting the potential for multi-exponential decay despite the diversity in pore sizes and impurity distributions within temperate glaciers. Similarly, we used a set of simplified one-dimensional water models despite the actual three-dimensional subsurface structure. All of these simplifications were necessary due to the high noise levels affecting our data, but this notwithstanding our approach was successful in constraining the range of possible water models.

From a practical standpoint, our methodology could be valuable for assessing natural hazards in glacial environments.

Although the total water volume estimated for Rhonegletscher is relatively small and located in a subglacial channel connected to the glacier portal, the approach could help in constraining englacial and subglacial water volumes for different, potentially hazardous settings. Future advancements in noise cancellation, survey strategies and instrumentation (e.g. Larsen et al., 2020; Grunewald et al., 2016) could enhance the utility of SNMR in glacier studies.

**Appendix A: Figures**





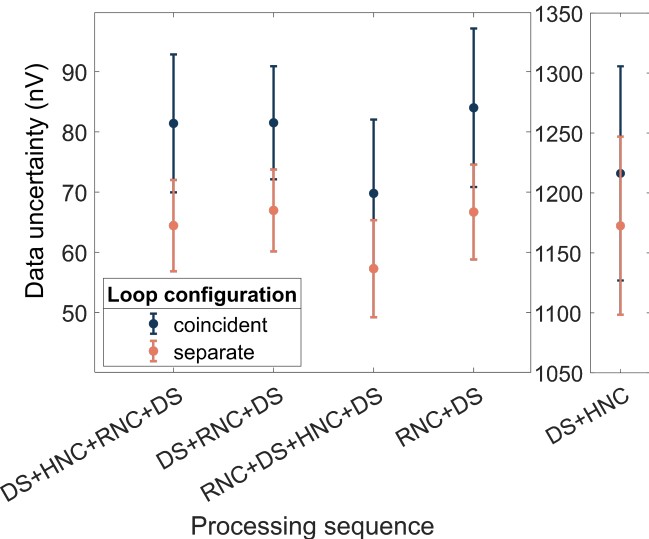

**Figure A1.** Noise remaining after different data-processing sequences. The data uncertainty of the coincident-loop (blue) and separate-loop (orange) measurements are shown separately. The error bars correspond to the standard deviation of the data uncertainties of different pulse moments about their mean value. Details on the computation of the data uncertainty can be found in Section 4.1. Abbreviations: DS = Despiking (parameters set in MRSmatlab: width - 10 ms, threshold - 5), HNC = Harmonic Noise Cancellation (parameters set in MRSmatlab: base frequencies - 50 Hz, 16.6 Hz), RNC = Remote Reference Noise Cancellation (parameters set in MRSmatlab: local transfer function)

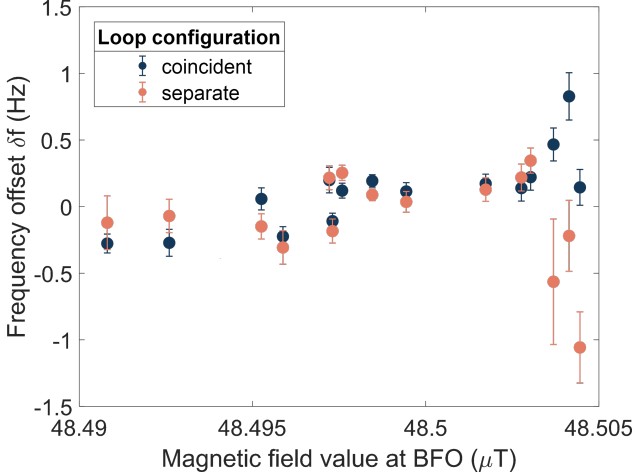

**Figure A2.** Correlation between the estimated frequency offset obtained from the SNMR measurement (y-axis) and the independently measured geomagnetic field at the Black Forest Observatory (BFO; x-axis) (Intermagnet). The coincident-loop (blue) and separate-loop (orange) measurements are shown separately.



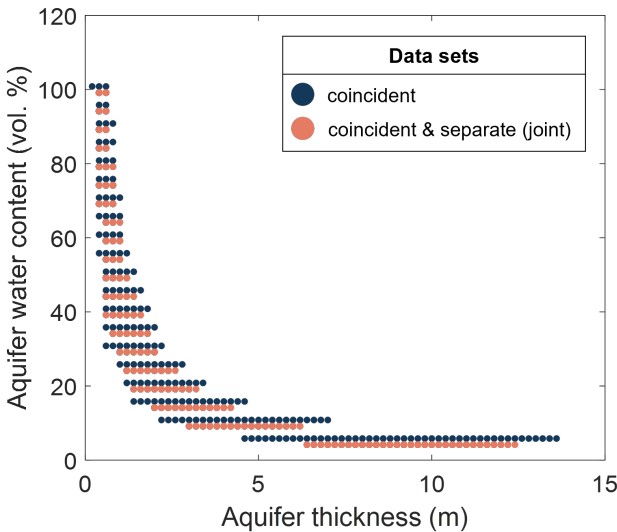

**Figure A3.** Correlation between the aquifer thickness (x-axis) and aquifer water content (y-axis) of compatible water models in the grid search (cf. Fig. 8).

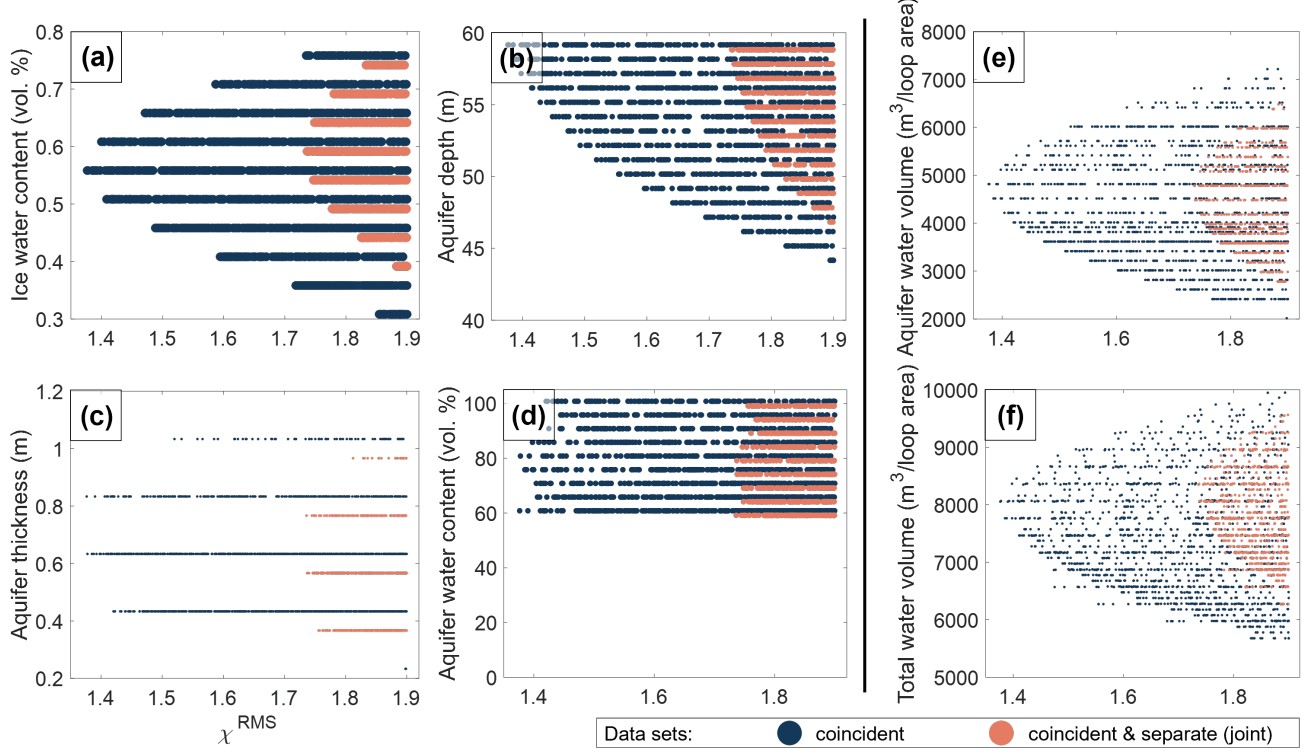

**Figure A4.** Same as Figure 8, but constraining the parameter space to models with an aquifer-water content $\geq 60\%$.



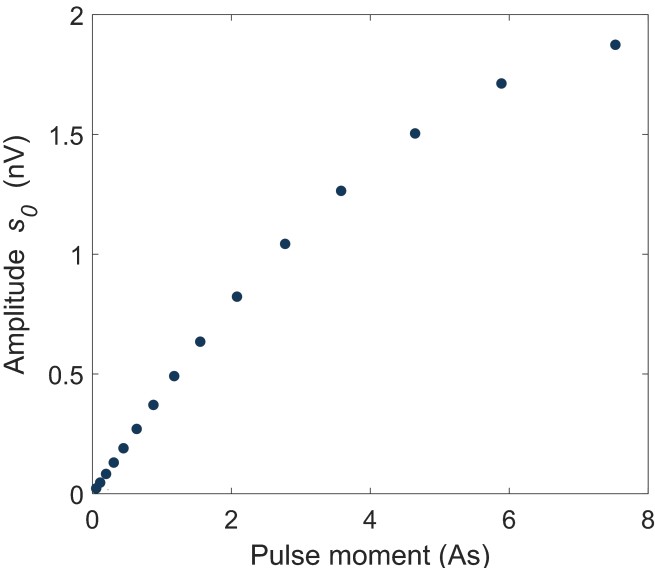

**Figure A5.** Synthetic sounding curve modelled for the far RNC loop (cf. Fig. 1b) assuming the three-layer model yielding the minimum $\chi^{\text{RMS}}$ in the coincident-loop analysis (cf. Tab. 2). For the modelling, we assumed a center-to-center distance between the Tx and RNC loops of $\approx 122\,\text{m}$.





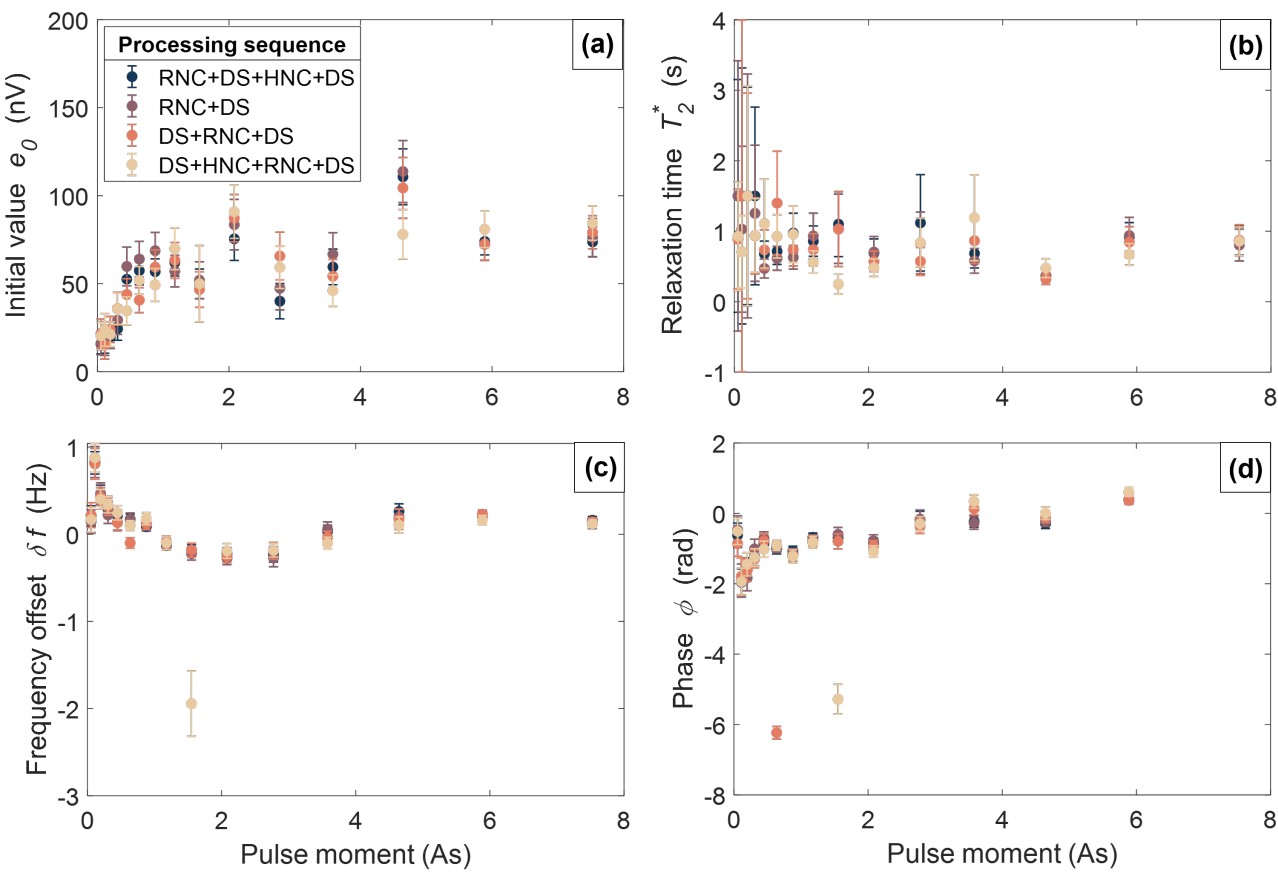

**Figure A6.** Estimation of the parameters from the mono-exponential fit $(e_0, T_2^*, \delta f, \phi)$ (cf. Eqs. 2 and 3) with corresponding uncertainties as a function of pulse moment. The different colours indicate the applied processing sequence. (a) Initial value $e_0$, (b) relaxation time $T_2^*$, (c) frequency offset $\delta f$, and (d) phase $\phi$.



*Code availability.* The code used for generating the results presented in this manuscript is available upon request. We plan to create a DOI that will include the relevant code upon acceptance of the manuscript.

*Data availability.* The raw data used in this study are available from the ETH online data repository polybox for the review process. Processed data can be obtained upon request. We plan to create a DOI that will include the raw data and any additional requested datasets upon acceptance of the manuscript.

*Author contributions.* MH, HM, DF and LG designed the SNMR survey on Rhonegletscher. MH, LG and RM planned and conducted the SNMR campaign. CO designed and carried out the GPR measurements at the same site. LG processed and analyzed the SNMR data with support from MH, MM, HM and DF. CO processed and interpreted the GPR data with the help of DF and HM. LG prepared the manuscript with the help of all co-authors.

*Competing interests.* Some authors are members of the editorial board of The Cryosphere.

*Acknowledgements.* This project was financially supported by the Swiss National Science Foundation (grant nr. 212061). The authors thank the following colleagues for their support during the fieldwork on Rhonegletscher in 2023: Janosch Beer, Leo Hösli, Filippo Ferrazzini, Cassandre Anthamatten, and Arabella Fristensky. Moreover, we want to thank Christoph Bärlocher for his technical support. We would like to thank the creators of MRSmatlab for developing, maintaining and sharing their software. Moreover, we thank Iris Instruments for their technical support with the Numis Poly instrument. The results presented in this paper benefited from data collected at magnetic observatories. We thank the national institutes that support them and INTERMAGNET (www.intermagnet.org) for promoting high standards of magnetic observations. We acknowledge the use of Grammarly in the final preparation of this article.



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
