# Peer review of "Surface nuclear magnetic resonance for studying an englacial channel on Rhonegletscher (Switzerland): Possibilities and limitations in a high-noise environment"

_EGUsphere, 2024_

## Author Comment (AC1)

**Authors' response to referee comment #1**

**RC** - Referee's comment

**AR** - Authors' response

**RM** - Revised manuscript

**RC** *The authors present a feasibility study for using SNMR to detect an englacial channel at the Rhonegletscher. The study focuses on the challenges encountered in a low signal, high noise environment and how to still image an englacial channel. The approach uses a grid search of different glacial water content models to try and fit the acquired data. The results are compared against a Ground penetrating radar (GPR) survey and show consistency between the two methods.*

**AR** We thank the referee for taking the time to provide us with such detailed feedback on our manuscript. In the section below, we address each comment individually and highlight the changes in the revised manuscript (underlined and blue).

**Main comments**

**Comment #1a**

**RC** *My comments relate mostly to the noise estimation and how the grid search is performed. It is mentioned that the average noise level is 70 nV for most pulse moments (P.12 L.268), but inspecting Figure 7, the largest error bar found here is ~18nV wide.*

**AR** It is correct that the average noise level amounts to 70 nV, while the error bars in Fig. 7 (or Fig. 6a) show significantly lower values. The two values are related to each other but they represent different types of uncertainties:
- Average noise level (see P.11, L. 249 – 254): The average noise level corresponds to the average data uncertainty. We calculate it by averaging all standard deviations $\sigma_D(q, t_i)$ that we compute for each time sample, pulse moment, and the real and imaginary parts of the time series.
- Error bars in Fig. 7 (see Section 3.2.1 and 4.1, specifically P.9, L.191 – 200, P. 12, L. 263 – 268): The error bars in Fig. 7a correspond to the standard deviation of the initial value $e_0(q)$. The initial value is derived from the four parameters $m(q)$ describing the mono-exponential decay. They are estimated with a least-squares approach. To assess the posterior uncertainties of the four model parameters, we compute the covariance matrix at the maximum-likelihood point $\tilde{C}_m \approx \left(G^T C_D^{-1} G\right)^{-1}$. $G$ is the linearised forward operator and $C_D$ corresponds to the data covariance, containing all data variances $\sigma_D(q, t_i)^2$. Ultimately, the standard deviation of the initial value (i.e. error bars in Fig. 7a) is retrieved from Eq. 4 in the manuscript, which

takes into account the extrapolation of the amplitude to earlier times (see. Eq. (3) in the manuscript).

We understand that the sentence on P. 12, L. 267, *"We observe amplitudes between 0 and 110 nV corresponding to roughly the order of magnitude of the average noise level (70 nV) for most pulse moments."*, might be misleading. In this sentence, we refer to the S/N, which we define as the ratio between the initial value and the average noise level.

**RM** In the revised version of the manuscript, we state the definition of the error bar in Figs. 6 and 7 more clearly in the corresponding caption. In addition, we highlight the difference between the average noise level and the error bars in Figs. 6 and 7:

*Figure 6. Estimation of the parameters from the mono-exponential fit (cf. Eqs. 2 and 3) with corresponding standard deviations (cf. Eq. 4 and the definition of the covariance matrix in Section 3.2.1) as a function of pulse moment. The coincident-loop data and the separate-loop data are shown in blue and orange, respectively. (a) Initial value $e_0$, (b) relaxation time $T_2^*$, (c) frequency offset $\delta f$, and (d) phase $\phi$.*

*Figure 7. Comparison of the measurements (dots with error bars, cf. Fig. 6a) and the synthetic sounding curves based on the minimum RMS-misfit models (lines) for the coincident-loop (blue) and the separate-loop configuration (orange). The different line types correspond to the three different models presented in Figure 4. (a) Comparison of the synthetic and measured sounding curve based on the coincident-loop configuration. (b) Comparison of the synthetic and measured sounding curve based on the coincident- and separate-loop configuration (joint data).*

Sec. 4.1, L. 268:

*Figure 6a depicts the estimated initial values $e_0(q_i)$ as a function of the pulse moments $q_i$ (sounding curve). We observe amplitudes between 0 and 110 nV corresponding to roughly the order of magnitude of the average noise level after processing $\sigma_D$ (70 nV) for most pulse moments. Note the difference in the magnitude between the average noise level $\sigma_D$ and the estimated standard deviation of the initial values $\sigma_{e_0}$ represented by the error bars in Fig. 6a. The two values are related to each other, but they represent different types of uncertainties (see Sec. 4.1 and 3.2.1 for their definition).*

**Comment #1b**

**RC** *In this figure (7), we see the forwarded data from three model scenarios having difficulties fitting the observed data within error bars. From one pulse moment to the next, the signal amplitude doubles and then drops by 35%, a difference way larger than the assigned error bars. Is the difficulty in fitting this data a product of the*

*simplified model scenario, or could it be a product of underestimating the uncertainty affecting the initial values?*

**AR** This is an excellent question and we see three possible explanations for the discrepancy between the fit and the data in Fig. 7:
Data:
- Misestimation of the initial values:
    - ○ Systematic misestimation as a result of processing (currently discussed in Sec. 5.4.1): RNC possibly distorts the signal up to 27 nV for the highest pulse moment (see L. 403). We expect a non-linear relationship between the amplitude of distortion and the pulse moment (see Fig. A5 and the fact that the phase of the transfer function does not have to be constant).
    - ○ Misestimation as a result of assuming a mono-exponential decay (currently discussed in Sec. 5.4.2, L.417 – 437): We fit the complex envelopes (Fig. 3) with mono-exponential decays (Eq. 2), assuming that all spins contributing to the signal exhibit similar relaxation times. This assumption might be inaccurate because the presence of impurities could lead to a multi-exponential decay.
- Misestimation of the standard deviation of the initial values (currently mentioned on P.9, L.191-200): When computing the standard deviations, we linearise the forward operator and assume normally distributed model parameters. Both assumptions might lead to a misestimation of the standard deviation.

Fit:
- Overly simplified water models (currently discussed in Sec. 5.4.2, L. 418 – 444): The one-dimensionality of the water models considered in this study is a significant simplification, since other studies like the GPR survey presented in Figs. 1b and 9 suggest a three-dimensional structure.

**RM** In the revised version of the manuscript, we adjusted and slightly rearranged the last paragraph of Sec. 4.2.1 (L. 302 - 306). We point to the relevant sections discussing the possible explanations mentioned above:

*If the synthetic data $e_0{}^{syn}(q_i)$ fit all of the observations $e_0(q_i)$ within their observational uncertainty $\sigma_{e_0(q_i)}$ , we expect $\chi^{RMS} \approx 1$. In our case, none of the models reach this value, suggesting a slight under-fitting. For instance, even the best model fails to replicate the amplitudes at lower pulse moments for the separate-loop data (Fig. 7b). This under-fitting is likely an expression of our simplified forward problem [cf. discussion in Sec. 5.4.2), a misestimation of the initial values $e_0(q_i)$ (cf. discussion in Sec. 5.4) or the initial values' uncertainties $\sigma_{e_0(q_i)}$ (limitations mentioned in Sec. 3.2.1).*

**Comment #2**

**RC** *Even with Equation 4 (P.9 L200), it is still unclear to me how the mean noise of 70nV becomes maximum 17nV in uncertainty on the model parameter e0. Please clarify.*

**AR** See response to comment #1a for the mathematical definition. We hope this sufficiently clarifies your question.

**RM** See response to comment #1a for how we clarified this in the text.

**Comment #3**
**RC** *In Figure 8b, the misfit for the models with a varying aquifer depth is shown. But unlike 8a, it seems it has not yet reached the lowest misfit, i.e., maybe an aquifer depth of 62m would be a better fit. Were the ranges chosen on previously acquired data (GPR)? If not, perhaps increasing the range here could reveal a similar parabola shape, like the one in Figure 8a.*

**AR** Yes, we chose the maximum depth based on the previously acquired GPR data, suggesting an average ice thickness of about 60 m in the survey area. Given the complexity of the aquifer geometry, e.g. non-horizontal ice surface and bedrock boundary, this is considered a robust estimate and thus did not investigate aquifer depths below that level.

**RM** In the revised version of the manuscript, we state more explicitly that we chose the maximum depth based on previously acquired GPR data (Sec. 3.2.2, L. 217-219):

*…The one-layer model consists of a 60 m thick, uniform ice column with a homogeneous liquid-water content (LWC) $x_{ice}$ (1 parameter). We chose the maximum depth based on the previously acquired GPR data, suggesting an average ice thickness of about 60 m in the survey area. …*

**Comment #4**

**RC** *These results of aquifer depth are later discussed (P.18 L. 359-361) as broadly consistent with the GPR profile which finds a channel at 40m. But the lowest misfit for the SNMR was with a channel at 59m depth.*

**AR** It is true that the lowest misfit is found for an aquifer at 59 m depth and that GPR measurements indicate a more shallow conduit. Since our one-dimensional water model is a simplification of reality and cannot represent a three-dimensional conduit, however, we are not surprised by this deviation from the GPR data.

**RM** In the revised version of the manuscript, we explicitly state that the minimum RMS-misfit model has a more shallow channel than indicated by GPR data (Sec. 5.2., L.360). We also extended the end of the section with a short discussion about the possible origin of this discrepancy (L.365):

*… From the GPR data, the average depth of the channel is around 40 m below the transmitter loop (Fig. 9). This is somewhat shallower than the minimum RMS-misfit model (59 m), but broadly consistent with the parameter distributions obtained from our SNMR investigations, which indicate a channel depth between 41 and 59 meters (Fig. 8b).*

*In addition to the channel, the GPR signals also reveal weak bedrock reflections and various features that we interpret as being part of the glacier's drainage system (including a surface water streams and possibly, a water-filled fracture; cf. Fig. 9). The spatial distribution of these partially englacial features indicates that our one-dimensional water models (cf. Fig. 4) might be an oversimplification as all of them have variable, three-dimensional shapes.*

*The simplified forward model could be the driver for the discrepancy between the aquifer depth of the minimum RMS-misfit model and the GPR findings. Additional factors may play a role too, such as signal distortions due to RNC, resulting in an overestimation of the aquifer depth. In Section 5.4, we further discuss the various limitations and their potential impact on the estimated model parameters.*

**Comment #5**

**RC**    *The RNC possibly distorting the signal up to 27nV is quite concerning since it is >25% of the maximum initial value seen (Figure 7). This is addressed in the conclusion, but only after stating that the RNC was the most crucial step in increasing S/N. Perhaps a more combined conclusion on RNC could highlight the usefulness and the issues with this approach.*

**AR**    We generally agree with the comment that it is important to jointly highlight the usefulness and issues of RNC, which we tried to implement with a different introduction in Sec. 5.4.2. However, to maintain the current structure of the discussion section, we decided to still focus on the limitations of RNC in Sec. 5.4.1.

**RM**    In the revised version of the manuscript, we now mention a possible distortion due to RNC already in Section 4.1; and start Section 5.4.1 by highlighting both the usefulness and the issues of RNC before providing more details.

      Sec. 4.1., L. 247 - 249

      *After processing the data according to the scheme in Figure 2, the signal-to-noise ratio of the time series increased significantly. While the application of DS and HNC slightly improved the S/N, the application of RNC was essential to reduce the noise level by an order of magnitude (Fig. A1). We note that noise cancellation with RNC has limitations due to possible distortions of the signal and discuss these in Sec. 5.4.1.*

      Sec. 5.4.1.

      *While RNC is the most crucial step in our noise-cancellation sequence, its usefulness is limited by its potential to distort the SNMR signal. In the following section, we attempt to estimate the effect of this distortion. For optimal noise cancellation, one wants to maximise the correlation between the time series of the remote reference loops and the receiver loop while detecting the SNMR signal exclusively in the receiver loop.*

**Comment #6**

**RC**    *Additionally, since a noise record has been recorded, would it be possible to use RNC on the noise only data and examine if the transfer functions are different? If they are different, it might be a sign of signal being distorted.*

**AR**    In principle, yes: If the noise was stable, we could have used a global transfer function. However, the noise conditions in this survey were not stable, and the transfer functions changed significantly between individual measurements. We thus used local transfer functions, i.e. transfer functions that are computed for each

recording. Against these unstable noise conditions, a more detailed analysis of the transfer functions was not considered useful.

**RM**    In the revised version of the manuscript, we add a sentence on the use of local transfer functions (Sec. 3.1.1., L. 142):

*3) Remote Reference Noise Cancellation (RNC) targets the noise of unknown characteristics, which is dominating our data. We deployed two remote reference loops to record the time series simultaneously with the two receiver loops (Fig. 1b). For this analysis, we only use the data from the loop further away to perform RNC, thereby reducing the amount of SNMR-signal contamination in the remote reference loop (see discussion in Section 5.4.1). To perform the cancellation and since the noise conditions were not stable, we used so-called local transfer functions, i.e. functions that are computed for each recording (Müller-Petke and Costabel, 2014).*

**Comment #7**

**RC**    *When assuming 100% water it vastly reduces the aquifer thicknesses found fitting data within the threshold. But is the instrument capable of resolving a <1m thick layer at 40m to 60m depth? Perhaps add some discussion on whether this is feasible given the selected pulse moments and loop dimensions.*

**AR**    We cannot resolve a 1 m layer at 40 – 60 m only using our SNMR data set. The data points in Fig. 8 should be seen as plausibility estimates for the respective model realisation. By using additional constraints based on assumptions or data from a different method, it is possible to further constrain the (water-model) parameter ranges. For example, in Fig. 8, we implicitly assume only positive water contents and set a maximum depth of 60 m (obtained from GPR data), which already constrains the ranges of the water-model parameters. In Fig. A4, we go one step further by assuming a minimum water content of 60% in the aquifer, which drastically reduces the estimated aquifer thickness to 1 m or thinner. Therefore, by adding further information (assuming a minimum water content of 60 %, because we expect a conduit mostly filled with water), the range of possible aquifer thickness was reduced to 1 m or less.

**RM**    In the revised version of the manuscript, we added a sentence to Sec. 5.1 on the resolution of thin layers (L. 349):

*…By doing so, the range of aquifer thicknesses decreases drastically, allowing for values between 0.2 and 1.0 meters. The ranges for the other parameters remain very similar to the ones in Figure 8. In conclusion, based on the information in Fig. 8 alone, we cannot resolve thin layers (≤ 1 m). However, by introducing additional information based on assumptions (e.g. minimum water content) or data from a different method (GPR in our case), it is possible to further constrain the range of the parameters in the water model – such as the aquifer thickness.*

**Comment #8**

**RC**    *A question about the englacial channel. I assume the water flowing within this channel, if so, how quickly? It might reduce the signal amplitude and should be discussed if appropriate.*

**AR**    It is correct that the water is generally flowing within subglacial channels, typically with velocities not exceeding 1m/sec (Werder et al., 2010 ). Flow typically has an impact on the estimates of relaxation time and water content in lab- or borehole-

NMR measurements, where the water molecules move through a heterogeneous magnetic (either due to an artificial magnetic field gradient in lab or logging measurements or a pore space gradient) field during the measurement (Callaghan, 1991). To be relevant, both the gradient and the flow need to be high. A high flow in a homogeneous field does not impact the measurement. Thus, we assume this effect to be negligible in our measurement, because of the presumably homogeneous Earth's magnetic field, and a small displacement of the molecules during the recording time (since the recording time is 1 sec and assuming the maximal flow velocity of 1m/s mentioned above, the expected maximal displacement is in the order of 1 meter). However, one would need to further investigate this effect to fully understand its impact in our case.

We decided to not include this into the discussion as we assume this to be a minor effect and very likely not relevant .

**RM** No changes are made in the revised version of the manuscript.

**Minor comments**

**Minor comment #1**

**RC** *P.5 L.108: The 16th q was not completed. Could this have helped constrain the aquifer depth in Figure 8 by increasing the depth of investigation?*

**AR** In general, yes, since the higher the pulse moment, the higher the sensitivity at deeper layers. However, due to time constraints, we were not able to repeat the incomplete measurement at the 16th q.

**RM** No changes are made in the revised version of the manuscript.

**Minor comment #2**

**RC** *P.12 L.261: The peaks at -20Hz are not seen in noise only spectrum in Figure 5b. Are these harmonics or related to transmitting at high pulse moments? And what harmonics do you expect at this frequency?*

**AR** The peaks at around -20 Hz are present in both spectra but less visible in the noise-only one. One reason for the difference is the visualisation is that the scale of Fig. 5b is slightly smaller than in Fig. 5a. In addition, this Figure is a 3D plot that was manually tilted. The tilting angle of the two figures might differ slightly, resulting in a different height of the peaks. Another reason could be processing: The presence of the SNMR signal in the data in Fig. 5a might have had an impact on the processing, i.e. the processing result might slightly differ for traces with and without SNMR signal.

We removed higher harmonics of ~16.6 Hz and ~50 Hz. The origin of the peaks at around -20 Hz is currently unknown. We suspect that the appearance at higher pulse moments is a temporal effect (i.e. source started emitting noise later in the day when the recordings at higher pulse moments occurred) and has no causal relationship with the pulse moment.

**RM**  We added a sentence in Sec. 4.1, L. 261:

> *The peaks at around 20 Hz indicate the presence of some residual higher harmonics that could not be removed with our processing routine. We suspect that the appearance of those peaks at higher pulse moments is a temporal effect (i.e. source started emitting noise later in the day when the recordings at higher pulse moments occurred) and has no causal relationship with the pulse moment.*

**Minor comment #3**

**RC**  *Figure 6a: Is it expected that the separate and coincident coil shows very different initial values? Is the water content lower here or is it mainly a product of less excitation?*

**AR**  Coincident- and separate-loop measurements have a different spatial sensitivity below the loops. Consequently, the so-called sounding curves show quite different shapes. Separated loops generally show smaller initial values $e_0$ readings at small pulse moments and larger initial values at higher pulse moments, when larger quantities of excited spins are excited below the receiver loop. The curves can have largely different shapes, depending on the size and direction of the loop offset (Hertrich et. al, 2005).

**RM**  We added a sentence on this effect in Sec. 4.2.1, L. 301:

> *…Based on these observations, we only consider three-layer models from now on. Note that the sounding curves of the separate- and coincident-loop data differ substantially (cf. Fig. 7b), which is a result of the difference in spatial sensitivity of the two configurations (Hertrich et al., 2005).*

**Minor comment #4**

**RC**  *Figure 8: Layout of figure is a bit confusing having the upper panel be (a),(b),(e), and the lower panel being (c),(d),(f). Perhaps consider three rows with a,b and c,d and lastly e,f..*

**AR**  We chose a more compact layout on purpose, since the figure would need a whole page otherwise. To avoid the confusion, we added a vertical line between the second and third column which apparently was not visible enough.

**RM**  In the revised version of the manuscript, we make the vertical line in Fig. 8 more visible.

**Minor comment #5**

**RC**  *Figure 9: Consider marking the maximum observed dimension of the englacial channel according to Church et al., 2021, if feasible.*

**AR** The maximum observed dimension of the englacial channel according to Church et. al., 2021, can be estimated from Fig. 1a. We judge that including this information also in Fig. 9 would not directly contribute to the key message of our paper and could confuse readers. Thus, we decided to leave Fig. 9 as it is.

**RM** No changes are made in the revised version of the manuscript.

**The following suggestions are directly implemented in the revised version of the manuscript.**

*P. 11 L.242: Indicate the abbreviation, i.e., "both the coincident(coi)- and separate(sep)-loop data…"*

*P20. L.426: a space missing between ",accumulate"*

*P.21 L. 464: A year is missing on the Ogier et al. reference.*

**References**

Callaghan, P. T.: Principles of Nuclear Magnetic Resonance Microscopy, Oxford University PressOxford, ISBN 97801985394459781383026610, https://doi.org/10.1093/oso/9780198539445.001.0001, 1991.

Hertrich, M., Braun, M., and Yaramanci, U.: Magnetic resonance soundings with separated transmitter and receiver loops, Near Surface Geophysics, 3, 141–154, https://doi.org/10.3997/1873-0604.2005010, 2005.

Müller-Petke, M. and Costabel, S.: Comparison and optimal parameter settings of reference-based harmonic noise cancellation in time and frequency domains for surface-NMR, Near Surface Geophysics, 12, 199–210, https://doi.org/10.3997/1873-0604.2013033, 2014.

Werder, M. A., Schuler, T. V., and Funk, M.: Short term variations of tracer transit speed on alpine glaciers, The Cryosphere, 4, 381–396, https://doi.org/10.5194/tc-4-381-2010, 2010.

---

## Author Comment (AC2)

**Authors' response to referee comment #2**

**RC** - Referee's comment

**AR** - Authors' response

**RM** - Revised manuscript

**RC** *In light of rapid glacier retreat and degradation of alpine permafrost, reliable observational methods for subsurface liquid water and ice contents are urgently needed. The authors present an interesting case study and a very rare application of surface nuclear magnetic resonance (SNMR) to detect and characterize an englacial channel of Rhonegletscher, Switzerland.*

*Although challenged by considerable and yet unknown sources of electromagnetic noise, the authors managed to derive a useable signal as well as plausible 1D models of the englacial hydrological setting by advanced data processing. Results were validated with their own colocated as well as previously acquired ground-penetrating radar measurements.*

*The manuscript is well structured and written with commendable scientific rigour reflected, among other things, by a discussion of alternative plausible models given the measurement uncertainty as well as a dedicated subsection to discuss the limitations of the approach presented.*

*I believe that the practical considerations and data processing steps presented here make a valuable contribution for researchers and practitioners applying SNMR in particular in, but not limited to, the emerging field of cryogeophysics.*

*The authors are kindly asked to consider the comments below in (minor) revisions of their exciting paper and excuse the delay in my review.*

*Kind regards*

*Florian Wagner*

*RWTH Aachen University*

**AC** We thank the referee for taking the time to provide us with such detailed feedback on our manuscript. In the section below, we address each comment individually and highlight the changes in the revised manuscript (underlined and blue).

**Main comments**

**Comment #1**

**RC** *1. State of the art: The authors rightfully state that application of SNMR in cryogeophysical settings is very rare and hence only a few studies are cited in the introduction together with some GPR studies. However, the cryogeophysical community has made substantial progress in recent years in quantifying subsurface liquid water and ice contents with other geophysical methods (e.g., electrical*

*resistivity tomography, seismic refraction, spectral induced polarization, etc.) also developing joint inversions to directly estimate water content by a combination of these methods. I feel the paper would benefit from acknowledging a few of these developments and properly placing the potential advantages and limitations of SNMR measurements in the overall effort of quantifying subsurface liquid water content with cryogeophysical methods.*

**AR**    We agree with the reviewer that there exist additional geophysical methods that can be used in cryospheric environments. However, in practice, those methods (ERT, SIP) cannot be meaningfully deployed on pure glacier ice (like Rhonegletscher), since the ground is too resistive. We added a short section on those methods in the reviewed version of the manuscript.

While electrical and electromagnetic methods have been successfully applied in geophysical applications in cryosphere studies in various settings (primarily in permafrost investigations), the investigation of pure temperate glacier ice usually shows resistivities in the MOhm range (Hochstein, 1967 and references of this study), which is too high to be investigated with electrical and electromagnetic techniques. Due to the small vertical velocity gradient in glacier ice, seismic refraction methods would be also unsuitable, but seismic reflection surveys proofed to be useful (Church et al., 2019).

**RM**    In the revised version of the manuscript, we added a sentence on the use of electrical and electromagnetic methods in glacierized environments and why their use on "pure" glacier ice to estimate the water content is limited (Sec. 1, L. 22):

*…While GPR and seismics are effective at detecting the boundaries of englacial structures, they do not provide direct information about water content in the ice, which can be of particular interest in the context of hazard management, like in the case of glacier water pocket outburst floods (Ogier et al.; Vincent et al., 2012; Haeberli, 1983).* *Although electrical and electromagnetic methods have been successfully applied in geophysical applications in cryosphere studies in various settings (primarily in permafrost investigations, e.g. Wagner et al., 2019; Mudler et al., 2022), the investigation of pure temperate glacier ice usually shows resistivities in the MΩ range (Hochstein, 1967), which is too high to be investigated with electrical and electromagnetic techniques.*

**Comment #2**

**RC**    *2. Inversion approaches: The main inversion is based on a grid search for the 1D water content distribution. Prior to this step, a least-squares inversion is conducted to fit the decay curves. I feel that these two inversions need to be separated more clearly. In particular, I find it confusing that for the first inversion a model vector is defined, but not for the actual inversion for layer thicknesses and water contents.*

**AR**    We agree that it makes sense to define two model vectors, one for the least-squares inversion and one for the grid search. We hope this helps to more clearly separate the two.

**RM** In the revised version of the manuscript, we added the definition of the water-model vector used in the grid search (Sec. 3.2.2, L. 228) and adapted the manuscript accordingly. We also changed the notation of the first model vector from $m$ to $m_1$:

*….We perform a grid search within the parameter space spanned by $m_2 = (x_{ice}, h_{aq}, d_{aq}, x_{aq}, h_{surf}, x_{surf})$ to identify the most likely water distributions f (z) explaining the measured e0(q). For all possible combinations of $m_2 = (x_{ice}, h_{aq}, d_{aq}, x_{aq}, h_{surf}, x_{surf})$, we repeat the following three steps (cf. Fig. 2):...*

**Comment #3**

**RC** *3. Noise discussion: The authors are very transparent about the poor data quality. However, the reader is kept left wondering where this noise comes from and how much is attributed to the (too close?) placing of the loops. Is it possible that the site is actually not that noisy, but the approach to estimate noise is not ideal?*

**AR** We are unsure if we understand this comment correctly. Typical anthropogenic noise sources in SNMR surveys include power lines, electric fences, and other electric infrastructure. On Rhonegletscher, we were not able to identify distinct sources of noise. Thus, we do not know the distance between the loops and the potential sources. We assume that the infrastructure in the larger area emitted electromagnetic waves that we recorded as noise. Presumably, in the highly resistive environment of crystalline rock and ice in the Rhonegletscher area, remote sources could have a stronger impact due to negligible electromagnetic attenuation. Possible sources are mentioned in more detail in Sec. 5.3, L. 376 - 380. Note that knowledge of noise sources and propagation is still missing in the field of SNMR.

**RM** We added a sentence on the possible effect of the highly resistive environment in Sec. 5.3., L379:

*Since no thunderstorms were recorded in the larger area during the survey either, we remain puzzled by the noise's origin. Presumably, in the highly resistive environment of crystalline rock and ice in the Rhonegletscher area, remote sources could have a stronger impact due to negligible electromagnetic attenuation. While the data exhibits some signatures of spikes and higher harmonics of 16.6 and 50\,Hz, the predominant noise is probably a superposition of multiple sources.*

**Comment #4**

**RC** *Also, care should be taken when comparing noise to other studies. For example, a link is made to a study in Denmark (Larsen and Behroozmand, 2016) where the magnitude of noise is compared on the basis of the RMS data misfit. To my understanding, the RMS misfit is a poor indicator for observational noise, as it can be dependent (and thus "tweaked") by the noise estimates, the quality of the forward model, the complexity of the subsurface parameterization, the inversion approach with its settings, and many other settings. In short, a "good" RMS misfit can also be obtained for a data set of poor quality or am I missing something here?*

**AR** The "RMS misfit" we mention on P.18, L. 372 is not the same as the RMS misfit defined in Eq. 7 on P. 11, otherwise, it would be a poor indicator indeed. The "RMS misfit" mentioned in the context of the study in Denmark has nothing to do with inversion fitting. Instead, it refers to the standard deviation of a time series with a mean value of zero (i.e. the predicted value for each time sample is zero).

Therefore, their "RMS misfit" is equivalent to the standard deviation we use to quantify the noise level. We understand that the wording is very misleading and will change that.

**RM**   In the revised version of the manuscript, we change the wording "RMS misfit" to "noise level" (P.18, L. 372):

… *For example, Larsen and Behroozmand (2016) studied the noise properties of multiple sites in Denmark. They investigated "sites with high-noise levels" showing* noise levels *of 0.25 and 0.3 nV m−2, which is almost one order of magnitude lower than the noise we recorded….*

**Comment #5**

**RC**   *4. Linguistic consistency: The authors currently mix British (e.g., "colours") and American English (e.g., "discretization"). Please choose one consistently throughout the paper. Additionally, I recommend to use the same term if the same thing is meant. For instance, "Earth's magnetic field" vs. "Earth's geomagnetic field" are both used in the paper. Choose one (or use the introduced B_earth symbol).*

**AR**   Thank you for paying attention to the linguistic consistency. We are changing the manuscript accordingly.

**RM**   In the revised version of the manuscript, we consistently use British English and use the same term for the same thing.

**Minor comments**

**Minor comment #1**

**RC**   *- L131-133: What is the despiking based on? A bit more information would be helpful here.*

**AR**   This processing step aims at eliminating spikes. MRSmatlab works with the so-called "TDmean" approach, where a spike in the single traces is identified if the amplitude is larger than a certain threshold. The segment with the spike in the single trace is then replaced by the stacked signal without the spike.

**RM**   *In the revised version of the manuscript, we added a sentence on despiking in Sec. 3.1.1, L. 131:*

*…Despiking (DS) removes extreme values (so-called spikes), like the one reaching more than 105 nV in Fig. 3a.* A spike is identified if the amplitude is larger than a certain threshold (typically set to five times the standard deviation of the time series, Mueller-Petke et al, 2016). The segment with the spike in the single trace is then replaced by the stacked signal without the spike. *Spikes are typically a result of*

*powerful discharges like lightning. While we identify multiple spikes in the data sets acquired on Rhonegletscher, they do not dominate the overall noise…*

**Minor comment #2**

**RC**   *- L141: Is the closer loop not helpful at all? Or could it be used to estimate the attenuation of the signal and hence the "objectivity" of the noise estimate somehow?*

**AR**   This is difficult to address in great detail. Primarily, the closer loop does not help because of even larger signal distortion, so we don't use it for processing. On the other hand, it also does not help for further noise investigations because it contains noise and signal. One could try to work on the noise-only records and study correlations. However, then we are running into the discussion of time-stability of the noise. So, yes one can think of "playing" with this data, but a discussion of this is beyond the scope of the paper.

**RM**   No changes are made in the revised version of the manuscript.

**Minor comment #3**

**RC**   *- L148: What exactly is meant by "best results"?*

**AR**   With "best results", we mean the lowest data uncertainty after processing.

**RM**   In the revised version of the manuscript, we properly explain what we replace "best results" with a more precise description (sec. 3.1.1, L. 148):

*…In Supplementary Fig. 1a, we compare the noise remaining after different processing sequences and show that the combination "RNC+DS+HNC+DS" is actually the one leading to the lowest remaining data uncertainty after processing.*

**Minor comment #4**

**RC**   *- L380: Could some of these noise sources be listed / discussed here?*

**AR**   See answer to comment #3.

**RM**   No changes are made in the revised version of the manuscript.

**The following suggestions are directly implemented in the revised version of the manuscript.**

*- L10: Maybe reformulate to "... consistent with simultaneously and previously acquired ground-penetrating radar measurements." (also to avoid the not yet introduced GPR abbreviation)*

*- L26 vs. L31: See general comment #4.*

*- L70, L91: The SNMR manufacturer is cited twice (with a URL and a not properly formatted*

*(?) citation in L91), whereas other manufacturers (e.g., Leica or Senors & Software) do not have a reference. I think manufacturers could be simply mentioned without website links, except for the loop recommendation in the manual, which needs to be properly formatted.*

*- L174: Use square brackets for the model vector here for better readability and consistency with the initial values in L179.*

*- Caption of Fig. 4 contains a mix of British and American English (see general comment #4)*

*- L232, L238: I think the introduction of an additional model vector here makes sense to avoid these repitions.*

*- L263: "provides" -> "provide" because this is referring to the "results", i.e. plural?*

*- L416 (and elsewhere): No hyphen between model and parameter needed.*

*- Both "Mueller-Petke" and "Müller-Petke" appear in the reference list.*

**References**

Church, G., Bauder, A., Grab, M., Rabenstein, L., Singh, S., and Maurer, H.: Detecting and characterising an englacial conduit network within a temperate Swiss glacier using active seismic, ground penetrating radar and borehole analysis, Annals of Glaciology, 60, 193–205, https://doi.org/10.1017/aog.2019.19, 2019.

Hochstein, M.: Electrical Resistivity Measurements on Ice Sheets, Journal of Glaciology, 6, 623–633, https://doi.org/10.3189/S0022143000019894, 1967.

Mudler, J., Hördt, A., Kreith, D., Sugand, M., Bazhin, K., Lebedeva, L., and Radi´c, T.: Broadband spectral induced polarization for the detection of Permafrost and an approach to ice content estimation – a case study from Yakutia, Russia, The Cryosphere, 16, 4727–4744, https://doi.org/10.5194/tc-16-4727-2022, 2022.

Müller-Petke, M., Braun, M., Hertrich, M., Costabel, S., and Walbrecker, J.: MRSmatlab — A software tool for processing, modeling, and inversion of magnetic resonance sounding data, GEOPHYSICS, 81, WB9–WB21, https://doi.org/10.1190/geo2015-0461.1, 2016.

Wagner, F. M., Mollaret, C., Günther, T., Kemna, A., and Hauck, C.: Quantitative imaging of water, ice and air in permafrost systems through petrophysical joint inversion of seismic refraction and electrical resistivity data, Geophysical Journal International, 219, 1866–1875, https://doi.org/10.1093/gji/ggz402, 2019.

---

## Author Response (AR2)

Dear Reinhard Drews,

We are pleased to hear that the paper has been accepted. We appreciate your patience throughout the peer-review process.

We have implemented your minor remarks:

- Removed "so-called" in most instances
- Added the ice-flow direction in Fig. 9
- Activated the DOI and updated the text of "Data availability"

Kind regards,

Laura Gabriel and co-authors